# WorldSimBench: Towards Video Generation Models as World Simulators

## Abstract

Recent advancements in predictive models have demonstrated exceptional capabilities in predicting the future state of objects and scenes. However, the lack of categorization based on inherent characteristics continues to hinder the progress of predictive model development. Additionally, existing benchmarks are unable to effectively evaluate higher-capability, highly embodied predictive models from an embodied perspective. In this work, we classify the functionalities of predictive models into a hierarchy and take the first step in evaluating World Simulators by proposing a dual evaluation framework called WorldSimBench. WorldSimBench includes **Explicit Perceptual Evaluation** and **Implicit Manipulative Evaluation**, encompassing human preference assessments from the visual perspective and action-level evaluations in embodied tasks, covering three representative embodied scenarios: Open-Ended Embodied Environment, Autonomous Driving, and Robot Manipulation. In the Explicit Perceptual Evaluation, we introduce the HF-Embodied Dataset, a video assessment dataset based on fine-grained human feedback, which we use to train a Human Preference Evaluator that aligns with human perception and explicitly assesses the visual fidelity of World Simulators. In the Implicit Manipulative Evaluation, we assess the video-action consistency of World Simulators by evaluating whether the generated situation-aware video can be accurately translated into the correct control signals in dynamic environments. Our comprehensive evaluation offers key insights that can drive further innovation in video generation models, positioning World Simulators as a pivotal advancement toward embodied artificial intelligence.

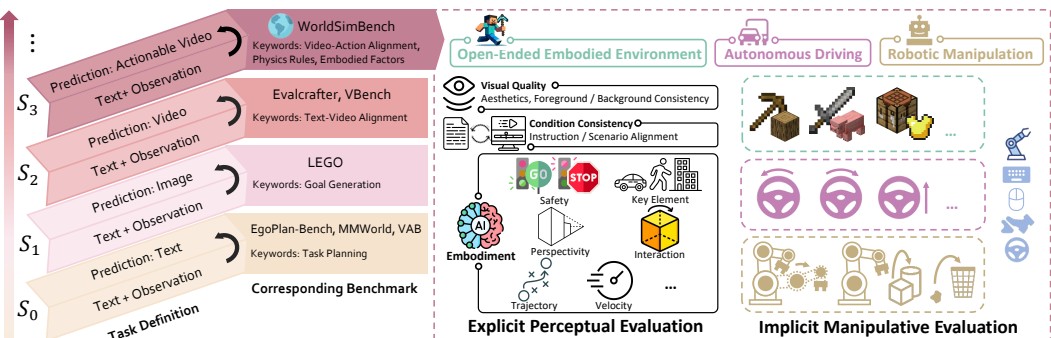

Figure 1: **Overview of the hierarchical capabilities of the Predictive Models.** Models at higher stages demonstrate more advanced capabilities. We take the initial step in evaluating Predictive Generative Models up to the $S_3$ stage, known as World Simulators, by introducing a parallel evaluation framework, WorldSimBench. WorldSimBench assesses the models both Explicit Perceptual Evaluation and Implicit Manipulative Evaluation, focusing on video generation and action transformation across three critical embodied scenarios.

Table 1: **Comparisons between existing Predictive Model benchmarks.** Interactive Environment refers to the interaction with the simulation environment during the prediction phase. Task-Level Interaction denotes that each task interacts once, whereas Action-Level Interaction represents the frequency of interactions that occur through the generation of actions for control purposes.

| Benchmark | Input Modality | Output Modality | Based Method | Stage | Interactive Env. | Evaluation Strategy |
|---|---|---|---|---|---|---|
| AgentBench (Liu et al., 2023b) | Text | Text | LLM | $S_0$ | Task-Level | Human Judgement |
| EgoPlan-Bench (Chen et al., 2023) | Text & Images | Text | MLLM | $S_0$ | N/A | Multi-choice |
| MMWorld (He et al., 2024) | Text & Images | Text | MLLM | $S_0$ | N/A | GPT Judgement |
| VAB (Liu et al., 2024a) | Text & Images | Text | MLLM | $S_0$ | Task-Level | Human Judgement |
| LEGO (Lai et al., 2023) | Text & Images | Image | IGM | $S_1$ | Task-Level | Feature Similarity |
| VBench (Huang et al., 2024) | Text | Video | VGM | $S_2$ | N/A | Feature Similarity |
| EvalCrafter (Liu et al., 2024b) | Text & Images | Video | VGM | $S_2$ | N/A | Feature Similarity |
| WorldSimBench | Text & Images | Actionable Video | VGM | $S_3$ | Action-Level | Human Preference Evaluator Embodied Metric |

# 1 INTRODUCTION

Before taking action, humans make predictions based on their objectives and observations of the current environment. These predictions manifest in various forms, *e.g.*, textual planning, visual imagination of future scene changes, or even subconscious planning at the action level. With the development of generative models, agents driven by these models are exhibiting predictive capabilities that enable them to complete embodied tasks by making human-like predictions, *e.g.*, high-level planning (Driess et al., 2023; Li et al., 2024), image-based guidance (Lai et al., 2023; Black et al., 2023), or future video prediction to drive actions (Du et al., 2023; 2024)). We refer to these models as **Predictive Models**. Recently, these models have been widely applied across various domains spanning from developing agents to solve inference tasks to leveraging predictions for driving robots to perform specific actions.

Nevertheless, the rich application scenarios and diverse model designs make predictive models a broad family. However, without categorizing them based on their inherent characteristics, the advancement of predictive model development remains limited. This leads to our first question: *Can we establish a reasonable hierarchical system for Predictive Models based on their output modality?* With a well-defined categorization, we can better target the evaluation of Predictive Models from different perspectives in diverse embodied environments, ensuring that their strengths and weaknesses are adequately assessed. In the literature, existing evaluations have typically focused on task planning capabilities by assessing text outputs or evaluating visual outputs from an aesthetic perspective. However, such approaches significantly limit the evaluation of highly embodied Predictive Models, as embodied scenarios are more concerned with physical properties (*e.g.*, perspective consistency, object breakability), which these methods fail to effectively assess. This brings us to our second question: *Can we conduct a more detailed evaluation of highly embodied Predictive Models from an embodied perspective?*

To answer the first question, we categorize the functionalities of Predictive Models into a hierarchy from $S_0$ to $S_3$, defined by the model's capabilities and output modality, accompanied by corresponding evaluation benchmarks as illustrated in Fig. 1. Models are classified based on the output modality in their output modalities. From lower to higher stages, the models are capable of generating: text, images, videos, and actionable videos (*i.e.*, the videos that can be translated into actions). It is worth noting that Predictive Models at $S_3$ capable of generating actionable videos integrate robust 3D scene understanding and physical rule priors to provide precise guidance for generating executable actions. These models are closely aligned with the recently proposed concept of World Simulators (Yang et al., 2023).

To answer the second question, we review the related benchmarks, as listed in Tab. 1. Evaluations on models in $S_0$ that generate text primarily focus on assessing task planning capabilities, while $S_1$ and $S_2$ assessments on visual output measure aesthetic quality through feature similarity analyses with ground truth data. With clearly defined evaluation dimensions and extensive annotated datasets, both types of assessments can be effectively conducted. However, evaluating World Simulators introduces complexities due to the intricate physical definitions involved. Additionally, conventional evaluation methods are inadequate for assessing the actionablilty of the generated videos, as there is no definite ground truth for actionable videos towards completing a specific embodied task. These factors pose significant challenges to the evaluation of World Simulators.

We argue that an evaluation aligned with human perception could provide a more intuitive and accurate reflection of the characteristics of the synthesized videos, including their adherence to physical rules. Besides, the actionability can be assessed through a closed-loop manner in simulations deployed with a unified video-to-action policy network. Considering these aspects, we take the very first step in evaluating World Simulators by proposing a dual evaluation framework called WorldSimBench. As shown in Fig. 1, WorldSimBench assesses World Simulators through two complementary approaches: **Explicit Perceptual Evaluation**, which focuses on the Visual Quality, Condition consistency, and Embodiment of the generated content, and **Implicit Manipulative Evaluation**, which measures the World Simulator's performance through the conversion of video into control signals. We present three representative embodied scenarios: Open-Ended Embodied Environment (OE), Autonomous Driving (AD), and Robot Manipulation (RM), to thoroughly evaluate the capability of World Simulators in generating and representing scenario-specific attributes.

In the Explicit Perceptual Evaluation, we first define evaluation criteria which is used to construct a comprehensive set of prompts specific to each scenario. The prompt lists are then used by various video generation models to produce a large number of video clips. Following extensive human feedback and annotation, these video clips are compiled into the HF-Embodied dataset which consists of a total of 35,701 tuples with multi-dimensional scores and fine-grained human feedback. Additionally, we train Human Preference Evaluator, using the HF-Embodied dataset to assess World Simulators at the perceptual level, offering a robust evaluation of both their visual fidelity and contextual accuracy. For the Implicit Manipulative Evaluation, we deploy three simulation environments for the three embodied scenarios respectively. These environments are used to collect data and train inverse dynamic or goal-based video-to-action models capable of mapping future videos to actions. In each of these embodied scenarios, the World Simulator is tasked with generating situation-aware videos in real-time, based on current observations and provided text instructions. These generated videos are then converted into actions using the pre-trained video-to-action models. The effectiveness of the World Simulator is implicitly evaluated by measuring the performance of the tasks, using relevant metrics to reflect the quality and accuracy of the generated video.

In summary, the main contributions are as follows: (1)We categorize the functionalities of Predictive Models into a hierarchy, defined by the model's capabilities and output modality, to advance research and development in the field and take the very first step in evaluating World Simulators. (2)We propose a dual evaluation framework called WorldSimBench, through Explicit Perceptual Evaluation and Implicit Manipulative Evaluation, we conducted a comprehensive evaluation of the World Simulator's capabilities from an embodied perspective, focusing on both the visual and action levels. (3)We conducted extensive testing across multiple models and performed a thorough analysis of the experimental results. Our findings highlight the strengths and limitations of current World Simulators and provide actionable insights for improving future video generation models. (4)We developed HF-Embodied Dataset, which includes fine-grained human feedback across three scenarios and 20 dimensions, with a total of 35,701entries. This dataset, containing both human ratings and the reasons behind them, not only enables the evaluation of World Simulators but also provides broader applications (*e.g.*,alignment) for future video generation models.

## 2 RELATED WORK

**Predictive Models.** Predictive models are capable of generating process representations that map the current state to future states by incorporating current state representations and control over future trends. Predictive Text Model, built on LLMs (Radford et al., 2019; Touvron et al., 2023; Chiang et al., 2023) and MLLMs (Achiam et al., 2023; Team et al., 2023; Liu et al., 2023a; Yin et al., 2023), generate future predictions in the text modality by accepting current state representations and text instructions. These models have demonstrated impressive performance in high-level planning tasks for embodied agents (Driess et al., 2023; Li et al., 2024; Qin et al., 2024; Chen et al., 2024; Zhang et al., 2024b; Lu et al., 2024). Similarly, image generation models (Brooks et al., 2023; Fu et al., 2023) as Predictive Image Model (Lai et al., 2023; Black et al., 2023; Zhou et al., 2024) can produce future goal images, showcasing strong capabilities during the decision-making phase of embodied agents. Predictive Video Model (Du et al., 2024; 2023), based on video generation models (Janner et al., 2022), have made some progress in embodied control. However, due to limitations in data or models, the generated videos often lack essential physical representations and logical consistency, restricting their applicability to fixed scenarios and single tasks.

With the advancement of diffusion transformer (Peebles & Xie, 2023) and the extensive utilization of large-scale internet video datasets (Bain et al., 2021; Ebert et al., 2021; Goyal et al., 2017; Grauman et al., 2022), certain Predictive Actionable Video Model (Yang et al., 2023) models, also known as World Simulators, have achieved more precise representations of physical laws and 3D environments.

**Evaluation of Predictive Models.** With the advancement of predictive models, research has also expanded to evaluate the capabilities of models at different stages. Liu et al. (2023b); Chen et al. (2023); Shi et al. (2024); Liu et al. (2024a) conducted text-level and task completion evaluations for Predictive Text Model at the $S_0$ stage. Lai et al. (2023) performed score-based evaluations from an aesthetic perspective for Predictive Image Model at the $S_1$ stage. Huang et al. (2024); Liu et al. (2024b) also assessed the aesthetic quality of videos generated by Predictive Video Model at the $S_2$ stage. We take the first step in evaluating World Simulators through an embodied perspective.

# 3 PREDICTIVE MODEL CATEGORY DEFINITION

In this section, we concretely categorize predictive models based on the model's capabilities and output modality. The detailed categorization stage of Fig. 1 is illustrated below,

• **Stage** $S_0$: At this stage, predictive models can generate corresponding predictions based on instructions and observations but are limited to textual modality. Benchmarks at this stage conduct text-level and task-completion evaluations through output text planning.

• **Stage** $S_1$: At this stage, predictive models can generate visual predictions based on instructions and observations, but without incorporating temporal information. Benchmarks at this stage conduct aesthetic evaluation for generated images.

• **Stage** $S_2$: At this stage, predictive models can generate corresponding video predictions based on both instructions and observations. Yet, due to limited model capabilities, the evaluation at this level focuses solely on the aesthetic quality of the generated outputs.

• **Stage** $S_3$: At this stage, predictive models can generate corresponding video predictions based on instructions and observations, with the predicted video content adhering to physical rules and aligning with the executed actions. These models are known as **World Simulators** (Ha & Schmidhuber, 2018; Yang et al., 2023), and WorldSimBench is a benchmark specifically designed to evaluate these World Simulators.

The rapidly evolving field of World Simulators offers exciting opportunities for advancing Artificial General Intelligence, with significant potential to enhance human productivity and creativity, especially in embodied intelligence. Therefore, conducting a comprehensive embodied evaluation of World Simulators is crucial.

# 4 WORLDSIMBENCH CONSTRUCTION

WorldSimBench evaluates the embodied capabilities of World Simulators across two distinct levels. The **Explicit Perceptual Evaluation** assesses the simulators based on human-perceived quality across different embodied scenarios, while the **Implicit Manipulative Evaluation** implicitly evaluates the simulators' capabilities by converting the generated videos into control signals and observing their performance in various closed-loop embodied tasks.

The evaluation of World Simulators encompasses three critical embodied scenarios: Open-Ended Embodied Environment (OE), Autonomous Driving (AD), and Robot Manipulation (RM). Minecraft serves as a popular testbed for OE, providing a challenging platform for agents to handle complex, unstructured tasks. In the context of AD, especially in outdoor settings, ensuring the stability and robustness of the agent's actions is crucial, making it an essential domain for assessing a World Simulator's capability in dynamic and uncertain environments. RM, a core task in embodied intelligence, demands precise and adaptive control, testing the world simulator's ability to generate actionable predictions that align with physical interactions. Together, these scenarios provide a comprehensive benchmark for evaluating the effectiveness of World Simulators across a range of real-world tasks.

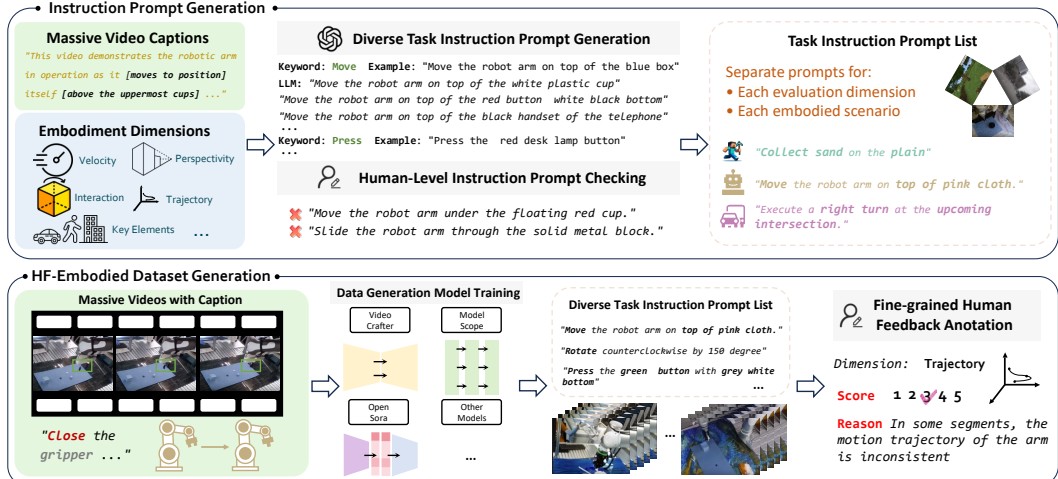

Figure 2: Overview of **Explicit Perceptual Evaluation**. (Top) **Instruction Prompt Generation.** We use a large collection of video captions from the internet and our predefined embodied evaluation dimensions. These are expanded using GPT and manually verified to create a corresponding Task Instruction Prompt List for data generation and evaluation. (Bottom) **HF-Embodied Dataset Generation.** Massive internet-sourced embodied videos with captions are used to train data generation models. Fine-grained Human Feedback Annotation is then applied to the embodied videos according to the corresponding Task Instruction Prompt List, covering multiple embodied dimensions.

## 4.1 EXPLICIT PERCEPTUAL EVALUATION

In Explicit Perceptual Evaluation, we propose Hierarchical Evaluation Dimensions, based on which we build a video assessment dataset annotated through fine-grained human feedback, named HF-Embodied Dataset. The dataset is constructed based on three key resources, each corresponding to a specific embodied scenario: a curated dataset of Minecraft videos from the internet for OE (Baker et al., 2022), real-world driving data for AD (Caesar et al., 2020), and real-world robot manipulation videos annotated with text instructions for RM (Chen et al., 2024). Using HF-Embodied Dataset, we train a Human Preference Evaluator to perform perceptual evaluations of World Simulators.

### 4.1.1 HIERARCHICAL EVALUATION DIMENSION

We develop a hierarchical evaluation dimension checklist for the three embodied scenarios, as illustrated in Tab. 2, which can be categorized into three main aspects: **Visual Quality**, **Condition Consistency**, and **Embodiment**. (1) Visual Quality primarily assesses the overall quality of video generation, including Aesthetics, Background and Foreground Consistency. (2) Condition Consistency focuses on the alignment with the input instruction. For tasks in OE that involve distinct scenarios, we additionally define Scenario Alignment to assess the alignment to the specific scenarios outlined in the instruction. (3) Embodiment has different definitions depending on the scenario. As all tasks require movement along a certain trajectory, we uniformly define Trajectory to evaluate the rationality of object movement in the video (*e.g.*, whether a robotic arm avoids obstacles during motion). In AD and RM, we define Perspectivity to assess whether the video exhibits a clear sense of depth. In OE and RM, we define Embodied Interaction to evaluate the plausibility of interactions with objects. We also define Velocity in OE to determine whether speed varies appropriately across different environments (*e.g.*, slower movement in water). In AD, we define Key Element to evaluate the rendering quality and consistency of crucial embodied elements, *e.g.*, pedestrians. We also introduce Safety in AD to assess whether the embodied actions comply with traffic rules. More details in Sup. A.

### 4.1.2 INSTRUCTION PROMPT GENERATION

Using the Hierarchical Evaluation Dimension and massive video captions from the key resources, we create a foundational but comprehensive prompt list. We utilize the knowledge of LLMs, *i.e.* ChatGPT, to extend the meta-prompts across a wide range. After manual screening for relevance, di-

Table 2: **Hierarchical Evaluation Dimension.** The dimensions are categorized into three main aspects: Visual Quality for evaluating the overall quality, Condition Consistency for evaluating the alignment to the input instruction, and Embodiment for evaluating embodied related factors like physical rules.

| Embodied Scenarios | Visual Quality | Condition Consistency | Embodiment |
|---|---|---|---|
| Open-Ended Embodied Environment (OE) | Background Consistency (BC) Foreground Consistency (FC) | Instruction Alignment (IA) Scenario Alignment (SA) | Velocity (VC) Trajectory (TJ) Embodied Interaction (EI) |
| Autonomous Driving (AD) | Aesthetics (AE) | Instruction Alignment (IA) | Perspectivity (PV) Trajectory (TJ) Key Element (KE) Safety (SF) |
| Robot Manipulation (RM) | Aesthetics (AE) Background Consistency (BC) Foreground Consistency (FC) | Instruction Alignment (IA) | Perspectivity (PV) Trajectory (TJ) Embodied Interaction (EI) |

versity, and data distribution, we compile the Task Instruction Prompt List, which separates prompts for each content-embodied scenario and each evaluation dimension, as shown in Fig. 2.

### 4.1.3 HF-EMBODIED DATASET GENERATION

**Data Preparation.** We select multiple video generation models and train them using a large corpus of videos and corresponding captions from the key resources. Due to the capabilities of the open-source video generation model, we conduct targeted training for each of the three distinct embodied scenarios individually, thereby developing several data generation models for different embodied scenarios. These models are then used to produce a substantial amount of instruction-following embodied videos, based on the corresponding captions, and the initial image condition where applicable (first frame conditioned text-to-video to generate situation-aware videos).

**Human Annotation.** We use human annotation to label the generated videos. Based on the Hierarchical Evaluation Dimension, we establish specific annotation guidelines and numerous in-conttext examples for the annotators. For each dimension, annotators are instructed to score the video solely based on its performance within that particular dimension and provide corresponding reasoning. For instance in RM, as illustrated in Fig. 2, under the dimension of Trajectory, annotators are required to evaluate the video exclusively on the generation quality of the motion trajectory. They are instructed not to consider other elements (*e.g.*, the rendering quality of the robot arm) or other dimensions (*e.g.*, consistency with instructions). Additionally, annotators are asked to provide fine-grained feedback on any deficiencies, *e.g.*, "inconsistent trajectory". As a result, we construct the HF-Embodied Dataset, which consists of a total of 35,701 tuples, each comprising a video, text instruction, multi-dimensional scores, and the potential reasons. More details in Sup. B.1.

### 4.1.4 HUMAN PREFERENCE EVALUATOR

The objective is to develop a video scoring model that assesses videos across multiple dimensions aligning with human perception. The model takes a generated video and a prompt as input and outputs a score ranging from 1 to $n$ ($n$ is defined specifically for each embodied scenario). The prompt includes both the video generation instructions and an explanation of the evaluation criteria. Leveraging the strong video understanding capabilities of multimodal large language models, we fine-tune Flash-VStream (Zhang et al., 2024a), a VideoLLM, aligning it with human perception on HF-Embodied Dataset. Only LoRA (Hu et al., 2021) parameters are trained. This enables the model to effectively grasp the evaluation metrics for embodied tasks and produce accurate scores, while maintaining its video perception and reasoning ability. We prove the effectiveness and generalizability of our Human Preference Evaluator in Sec. 5.2.

### 4.1.5 EVALUATION METRICS.

The evaluation of a video generation model is based on the scores assigned by the evaluator across various dimensions. For each dimension, the video generation model generates videos guided by several carefully selected instructions sourced from Task Instruction Prompt List that are strongly aligned with the specific evaluation criteria, *e.g.*, "explore on the beach" for Embodied Scenario in OE. The final metric for each model is computed as the average score across all dimensions. The evaluated dimensions for each embodied scenario are listed in Tab. 2.

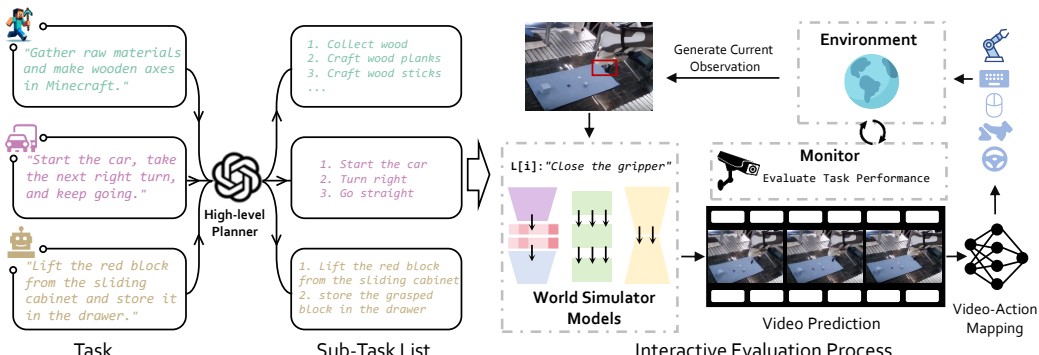

Figure 3: Overview of **Implicit Manipulative Evaluation**. Embodied tasks in different scenarios are decomposed into executable sub-tasks. The video generation model generates corresponding predicted videos based on the current instructions and real-time observations. Using a pre-trained IDM or a goal-based policy, the agent executes the generated sequence of actions. After a fixed timestep, the predicted video is refreshed by sampling again from the video generation model, and this process repeats. Finally, the success rates of various embodied tasks are obtained through monitors in the simulation environment.

## 4.2 IMPLICIT MANIPULATIVE EVALUATION

The Implicit Manipulative Evaluation assesses the capabilities of World Simulators across various embodied scenarios by treating the simulator as a low-level decision maker for situational contexts. Using pre-trained video-to-action models, we implicitly evaluate the performance of the World Simulators by observing their effectiveness in closed-loop embodied task tests.

### 4.2.1 SIMULATION CONSTRUCTION

The Implicit Manipulative Evaluation is conducted using the following three simulation platforms, for specific settings, please refer to the Supplementary Material.

OE We employ MineRL as the Minecraft simulator, with the observation space limited to RGB images and the action space confined to keyboard and mouse controls. We adopt the Steve-1 benchmarks (Lifshitz et al., 2024), with task descriptions *e.g.*, "chop a tree."

AD We conduct standard closed-loop evaluations using the CARLA (Dosovitskiy et al., 2017) simulator on the LangAuto Benchmark (Shao et al., 2024). Task descriptions include instructions like "do not deviate from this route."

RM We employ CALVIN (Mees et al., 2022) as the robot manipulation simulator, using only RGB images for the observation space and limiting the action space to the 7-DOF (degrees of freedom) of the robot arm. Task descriptions include commands *e.g.*, "pull the handle to open the drawer."

### 4.2.2 EMBODIED TASK EVALUATION

**Evaluation Pipeline.** As illustrated in Fig. 3, we first leverage existing or custom-trained video-to-action models as intermediaries between the World Simulator and the agent performing closed-loop tasks, for the selected benchmarks across three simulation environments. This approach enables the transformation of the predicted future videos from the World Simulator into executable control signals in real-time, thereby indirectly evaluating the World Simulator's capability through the successful completion of embodied tasks. The evaluation process is tailored to the specific nature of the models under consideration, establishing distinct protocols for closed-loop task evaluation. We fine-tune the models on simulation datasets tailored to each task. These datasets, derived from the three aforementioned benchmarks, include task instructions and corresponding videos, ensuring the models are well-adapted to the specific embodied scenarios. Finally, the evaluated World Simulator is integrated with the video-to-action model to jointly form an embodied agent that performs the given tasks. The agent's performance across various tasks serves as a direct measure of the World Simulator's effectiveness.

Table 3: **The overall performance comparison between Human Preference Evaluator and GPT-4o.** HPE indicates Human Preference Evaluator. HPE@Lavie means that HPE is trained on videos except those generated by Lavie. The validation is conducted on videos generated by Laive under zero-shot setting.

| Embodied Scenario | GPT-4o | HPE | GPT-4o@OpenSora | HPE@OpenSora | GPT-4o@Lavie | HPE@Lavie |
|---|---|---|---|---|---|---|
| OE@Acc($\uparrow$) | 72.8 | **89.4** | 66.5 | **71.6** | 78.5 | **87.9** |
| AD@PLCC($\uparrow$) | 0.28 | **0.60** | 0.03 | **0.34** | -0.04 | **0.49** |
| RM@PLCC($\uparrow$) | 0.07 | **0.43** | -0.06 | **0.47** | 0.17 | **0.44** |

**Evaluation Metrics.** In OE, we track the MineRL (Guss et al., 2019) environment state to calculate metrics *e.g.*, travel distance and early-game item collection. Travel distance is the agent's maximum horizontal displacement (X-Z plane) from the spawn point, while dig depth is its maximum vertical displacement (Y axis). We record the maximum number of logs, seeds, and dirt items in the agent's inventory during the episode. In AD, we employ eight widely used evaluation metrics in Carla (Dosovitskiy et al., 2017), including Route Completion (RC), Infraction Score (IS), Driving Score (DS), Vehicle Collisions (VC), Pedestrian Collisions (PC), Layout Collisions (LC), Red Light Violations (RV), and Offroad Infractions (OI). In RM, we evaluate the video generation model in the CALVIN (Mees et al., 2022) setting (train on A, B, C $\rightarrow$ test on D) by running 20 trials and calculating the average success rate.

## 5 EXPERIMENTS

### 5.1 EXPERIMENTAL SETUP

We evaluate 8 popular video generation models, including Open-Sora-Plan(T2V) (Lab & etc., 2024), Lavie (Wang et al., 2023c), ModelScope (Wang et al., 2023b), OpenSora (Zheng et al., 2024), AnimateDiff (Guo et al., 2023), Open-Sora-Plan(TI2V) (Lab & etc., 2024), Dynamicrafter (Xing et al., 2023), EasyAnimate (Xu et al., 2024) through both Explicit Perceptual Evaluation and Implicit Manipulative Evaluation, across three distinct scenarios: Open-Ended Embodied Environment (OE), Autonomous Driving (AD), and Robot Manipulation (RM). All models finetuned on specific datasets corresponding to three embodied scenarios in Explicit Perceptual Evaluation and Implicit Manipulative Evaluation. Detailed information on the datasets, training, and testing configurations can be found in the Supplementary Material.

For Explicit Perceptual Evaluation, we extract five instructions from the Task Instruction Prompt List for each dimension across the three embodied scenarios, ensuring they strongly align with the specific evaluation criteria, as discussed in Sec. 4.1.5. The selected instruction prompts each model to generate five videos, which are then scored by the Human Preference Evaluator to obtain an average score for the model's performance. For the scoring range 1-$n$, $n$ is set 2 for OE, and set 5 for both AD and RM. We indicate that the generation quality in OE is perceived as binary from a human perspective, while the other two scenarios exhibit a more diverse range of video quality.

For Implicit Manipulative Evaluation, we constructed three video-to-action models for embodied simulation environments, following the designs of Steve-1 (Lifshitz et al., 2024), Susie (Black et al., 2023), and LMdrive (Shao et al., 2024). For the evaluated models, we used the following datasets for fine-tuning: (1) VPT (Baker et al., 2022) and our own collected videos along with corresponding task descriptions as the training set for the OE; (2) the full Calvin(ABC_D) (Mees et al., 2022) video dataset and corresponding robot arm control instructions as the training set for RM; and (3) the full Carla (Dosovitskiy et al., 2017) video dataset and corresponding autonomous driving navigation commands as the training set for AD. Since the video-to-action model in our OE setup utilizes a goal-based policy, which interprets the goal from the input video and generates actions based on the current observations and the goal, it allows us to additionally evaluate text-to-video models.

### 5.2 EXPERIMENTS ON HUMAN PREFERENCE EVALUATOR

We demonstrate the strong capabilities and generalization of Human Preference Evaluator by comparing it with GPT-4o (OpenAI, 2024), showcasing its applicability for Explicit Perceptual Evaluation, as shown in Tab. 3. We use accuracy (Acc) in OE to assess the alignment of the model with

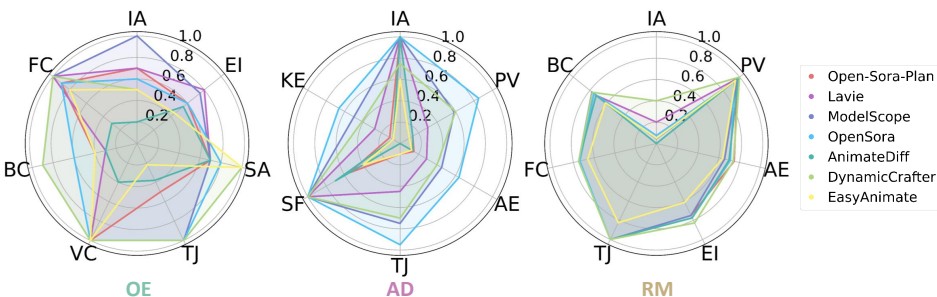

Figure 4: **Result of Explicit Perceptual Evaluation aross three embodied scenarios.** Scores in each embodied scenario are normalized to 0-1. The abbreviations are listed in Tab. 2.

human preferences, given the scoring range of 1-2. In contrast, we employ Pearson linear correlation coefficient (PLCC) for AD and RM as their scores range from 1-5.

After fine-tuning on HF-Embodied Dataset, our evaluator consistently surpasses the performance of GPT-4o in terms of alignment with human preferences across all scenarios. Additionally, we conducted zero-shot experiments with two challenging models, *i.e.* OpenSora and Lavie. GPT-4o exhibits a negative correlation with human preferences in evaluating OpenSora in AD under zero-shot setting, as well as evaluating Lavie in RM under zero-shot setting. Our evaluator's zero-shot performance shows a high correlation with human preferences, further demonstrating its robust generalization capabilities. Human Preference Evaluator is suitable for Explicit Perceptual Evaluation, and the HF-Embodied Dataset can be leveraged to train even more aligned models for assessing video generation models towards World Simulators. More details in Sup. B.3.

## 5.3 DESIGN FEATURES AND DISCUSSIONS

In this section, we discuss the Design features and corresponding observations we draw from our comprehensive evaluation experiments. More details can be found in the Supplementary Material.

**Human Prefrence with Feedback.** Given the complexity and diversity in the representation of physical rules in videos, even a specific dimension may manifest in various ways (for example, both illogical and discontinuous object motion fall under trajectory-related issues). This makes it challenging to evaluate using score-based models or a single fixed set of evaluation criteria. WorldSim-Bench addresses this challenge effectively by employing a human preference scoring mechanism and a fine-grained feedback system. Fig. 4 illustrates the evaluation results of Explicit Perceptual Evaluation, more detail analyze could be found in Sup. C. In OE, most models struggle with Embodied Interaction, particularly in generating plausible object deformations, *e.g.*, block shattering, due to the complexity of physical rules. In AD, the variation between models is minimal, with high-performing models excelling across all dimensions. The simpler instructions, like moving forward or turning, lead to high Instruction Alignment, but many generated videos suffer from poor 3D depth (Perspectivity) and fail to depict realistic embodied elements like pedestrians and vehicles, affecting the overall Aesthetic. In RM, models perform uniformly well in static scene depiction, excelling in Perspectivity and Foreground/Background Consistency. However, they struggle with Instruction Alignment, often generating aimless actions. Despite this, the lack of unreasonable trajectories results in relatively high Trajectory scores, though robotic manipulation remains a significant challenge for current models.

**Close-loop Interactive Evaluation.** Given the dynamic nature and real-time requirements of interactive environments, evaluating World Simulators through static benchmarks often fails to capture the full spectrum of their capabilities. Close-loop Interactive Evaluation addresses this by enabling continuous feedback and adaptation, ensuring that the model's predictions and actions evolve in response to the changing environment, thus providing a more accurate and realistic assessment of its performance. Fig. 5 presents the Implicit Manipulative Evaluationevaluation results, showing significant variation in the performance of video generation models across different tasks. In the OE, video generation models conditioned on the first frame have a significantly lower success rate compared to those without image conditioning. This suggests that models with image conditioning struggle to generate physical laws and 3D scene representations accurately. Tasks like travel, requiring high-quality trajectories and 3D representation, show the greatest variation in model performance, while

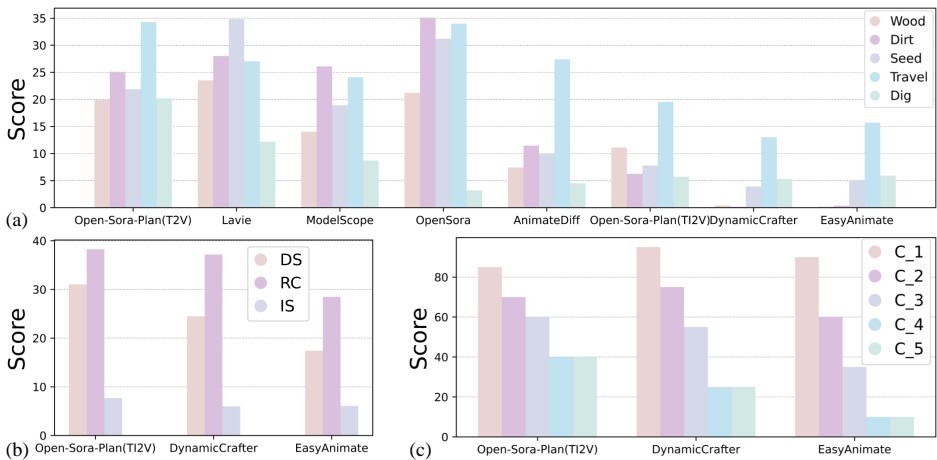

Figure 5: **Result of Implicit Manipulative Evaluation aross three embodied scenarios.**The abbreviations are listed in Sec. 4.2.2.

simpler tasks like collecting wood see similar performance across models, indicating effective handling of minimal background variation. In the AD, models with better trajectory(Open-Sora-Plan) generation perform better. In the RM, where background variation is minimal, models perform similarly on simple tasks, but as complexity increases, more robust models achieve higher success rates. Despite some success across scenarios, video generation models still need significant improvements in generating physically consistent content to be reliable for training agents or guiding actions.

**Alignment of Physical Rules and Actions.** Ensuring that World Simulators adhere to physical laws while generating predictions is crucial for practical application. The alignment of physical rules and actions is essential as it guarantees that the model's outputs are not only visually plausible but also executable in real-world scenarios. This approach allows for the seamless integration of predicted actions with their physical environment, ensuring reliability and effectiveness in real-world tasks. Based on our experimental findings, we observe that most conclusions from the Explicit Perceptual Evaluationand Implicit Manipulative Evaluationevaluations are consistent. Specifically, the visual quality across most dimensions aligns with the results from the closed-loop experiments. *e.g.*, Dynamicrafter, which performs well in trajectory generation in Explicit Perceptual Evaluation, also excels in trajectory-focused scenarios like AD and RM. However, in other cases—such as the OE, which requires more frequent interactions, and long-sequence tasks (4, 5) in RM—Dynamicrafter underperforms compared to Open-Sora-Plan. This differs from the Explicit Perceptual Evaluation results, likely because these tasks demand stable, high-quality video generation for guidance, where Open-Sora-Plan shows higher robustness. Therefore, a comprehensive evaluation of video generation models requires a combination of Explicit Perceptual Evaluation and Implicit Manipulative Evaluation assessments to provide the most fair and accurate judgment. Finally, based on the overall Explicit Perceptual Evaluationand Implicit Manipulative Evaluationresults, we conclude that current video generation models still fail to effectively capture many physical rules, indicating significant improvements are needed before they can function as true World Simulators.

## 6 CONCLUSION

In this work, we classify the functionalities of predictive models into a hierarchy and take the first step in evaluating World Simulators by proposing a dual evaluation framework called WorldSim-Bench. We conducted a comprehensive evaluation and analysis of multiple video generation models as World Simulators through both Explicit Perceptual Evaluation and Implicit Manipulative Evaluation processes. We summarize key findings from the evaluation and hope these insights will inspire and guide future research on World Simulators.

**Limitations.** Although we evaluate physical rules and 3D content from the perspective of embodied intelligence, the World Simulator can be applied to more scenarios than just robots, and different scenarios have more physical representations, so how to effectively evaluate the World Simulator in other scenarios requires more exploration.

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

# Part I

# Appendix

## Table of Contents

## A    TAXONOMY IN EXPLICIT PERCEPTUAL EVALUATION

We outline the evaluation dimensions for each embodied scenario below, along with their corresponding explanations. These explanations are used for detailed human annotation documentation and also serve as the explanation of the evaluation criteria in instructions for the Human Preference Evaluator.

### A.1    OPEN-ENDED EMBODIED ENVIRONMENT

**Visual Quality.** Background Consistency ensures the background remains consistent throughout the video. Foreground Consistency verifies the consistency of the foreground elements.

**Condition Consistency.** Instruction Alignment assesses whether the video aligns with the provided input instruction. Scenario Alignment checks if the input instruction defines an embodied scenario and whether the video accurately reflects this scenario.

**Embodiment.** Velocity evaluates if the velocity of the observed object is appropriate. Embodied Interaction evaluates the embodied interaction's appropriateness based on the interaction process and target. Trajectory evaluates whether the motion trajectory in the video is logical.

### A.2    AUTONOMOUS DRIVING

**Visual Quality.** Aesthetics evaluates whether the composition, color, lighting, and scene in the video align with human aesthetics.

**Condition Consistency.** Instruction Alignment assesses whether the video aligns with the provided input instruction.

**Embodiment.** Perspectivity evaluates the video's perspective, specifically assessing the 3D scene relationships. This includes evaluating whether the video has a strong sense of depth and realism (*i.e.*, whether it feels three-dimensional). Additionally, assess the logic of lighting and shadows, including whether the shadow positions are consistent with the light sources. Trajectory evaluates whether the movement and the trajectory of elements in the video is logical. Key Element assesses the generated quality of embodied elements *e.g.*, roads, vehicles, pedestrians, bicycles, lane markings, sidewalks, traffic signs, and traffic lights. Safety evaluates whether the behavior of the vehicles comply with traffic rules. Are there any instances of running red lights, speeding, or driving outside of permissible areas.

### A.3    ROBOT MANIPULATION

**Visual Quality.** Aesthetics evaluates whether the composition, color, lighting, and scene in the video align with human aesthetics. Background Consistency ensures the background remains consistent throughout the video, include the manipulation table and the environment. Foreground Consistency verifies the consistency of the foreground elements, including the robotic arm and the object on the manipulation table.

**Condition Consistency.** Instruction Alignment assesses whether the action of the robot arm in the generated video aligns with the provided input instruction.

**Embodiment.** Perspectivity evaluates the video's perspective, specifically assessing the 3D scene relationships. This includes evaluating whether the video has a strong sense of depth and realism (*i.e.*., whether it feels three-dimensional). Additionally, assess the logic of lighting and shadows, including whether the shadow positions are consistent with the light sources. Embodied Interaction judges whether the object's shape and posture conform to the rules during the collision of objects and the interaction between the robotic arm and the object. Trajectory evaluates whether the trajectory of the robotic arm is reasonable and in line with human cognition.

Table 4: **Analysis of HF-Embodied Dataset.** Samples scored higher than 3 in AD and RM are considered positive.

| Embodied Scenario | #instructions | #videos | #dims | #actions | #positive | #negative |
|---|---|---|---|---|---|---|
| Open-Ended Embodied Environment | 270 | 8401 | 7 | 11 | 121249 | 79965 |
| Autonomous Driving | 5 | 15870 | 6 | 5 | 56768 | 35044 |
| Robot Manipulation | 2556 | 11430 | 7 | 26 | 70672 | 9338 |

# B  DETAILD IMPLEMENTATION OF EXPLICIT PERCEPTUAL EVALUATION

## B.1  HF-EMBODIED DATASET

Tab. 4 provides an analysis of the HF-Embodied Dataset. In Autonomous Driving scenario, there are only five instructions: move forward, move backward, turn left, turn right, and stop. The other two scenarios include a variety of instructions that combine actions with target objects. Given the diverse instructions, different video generation models generate multiple videos after finetuning on specific datasets. To enhance the Human Preference Evaluator understanding of the autonomous driving context, we also supplement the AD scenario with videos from real-world scenes. Additionally, we list the quantities of positive and negative samples across all dimensions. Samples with human annotated scores of 3 or higher in AD and RM are considered positive. Leveraging HF-Embodied Dataset with comprehensive embodied dimensions, we train the Human Preference Evaluator to enable efficient assessment in Explicit Perceptual Evaluation.

**Discussion of Future Work.** Human Preference Evaluator (HPE) and HF-Embodied Dataset have been effective in aligning generated content with human preferences and evaluating video generation models, and we could explore more about its potential applications. Here are some future work directions to leverage the capabilities of HPE and HF-Embodied Dataset:

**Interactive Training for Generative Models** Utilize HPE as a real-time feedback mechanism during the training of generative models. By integrating HPE and HF-Embodied Dataset into a reinforcement learning framework, it could dynamically guide the model to improve alignment with human preferences across various scenarios, and can even make the world simulator perform better in downstream tasks.

## B.2  VIDEO GENERATION MODEL FINETUNING

Table 5: **Training Frames of Generation Models.**

| Model | Open-Sora-Plan | Lavie | ModelScope | OpenSora | AnimateDiff | DynamicCrafter | EasyAnimate |
|---|---|---|---|---|---|---|---|
| Short Videos(frames) | 16 | 16 | 16 | 16 | 16 | 16 | 16 |
| Long Videos(frames) | 64 | 48 | 60 | 48 | 64 | 60 | 64 |

We evaluate 8 popular video generation model, including Open-Sora-Plan(T2V) (Lab & etc., 2024), Lavie (Wang et al., 2023c), ModelScope (Wang et al., 2023b), OpenSora (Zheng et al., 2024), AnimateDiff (Guo et al., 2023), Open-Sora-Plan(TI2V) (Lab & etc., 2024), DynamicCrafter (Xing et al., 2023), EasyAnimate (Xu et al., 2024) through both Explicit Perceptual Evaluation and Implicit Manipulative Evaluation, across three distinct scenarios: Open-Ended Embodied Environment (OE), Autonomous Driving (AD), and Robot Manipulation (RM).

In Open-Ended Embodied Environment, we use **OpenAI Contractor Gameplay Dataset** (Baker et al., 2022) which is created by hiring human contractors to play Minecraft and complete tasks like house building. Keypresses and mouse movements are recorded during gameplay. We apply the same preprocessing steps as VPT, including filtering out null actions. Additionally, we create a supplementary dataset for the task "Explore" by generating trajectories using various pre-trained Steve-1 agents. The distribution of this dataset is enhanced by randomly switching between models during trajectories, resetting the agent's memory, and adjusting the agent's orientation to face new directions at random intervals. For specific in-game events, *e.g.*, "mine_block", the type of block broken is logged alongside precise timestamps. These timestamps allow for accurate progress tracking and are aligned with the completion of event-related instructions.

| OE@Acc(↑) | BC | FC | IA | SA | VC | TJ | EI | Overall |
|---|---|---|---|---|---|---|---|---|
| GPT-4o | 60.5 | 70.4 | 70.9 | 67.3 | 79.6 | 83.7 | 85.9 | 72.8 |
| HPE | **81.2** | **87.5** | **87.5** | **96.4** | **94.5** | **93.8** | **88.8** | **89.4** |
| GPT-4o@OpenSora | 60 | 80 | **80** | 50 | 0.0 | **100** | **88.8** | 66.5 |
| HPE@OpenSora | **70** | **90** | 60 | **100** | **100** | 22.2 | 80 | **71.6** |
| GPT-4o@Lavie | 50 | 66.7 | 75 | 88.8 | 87.5 | **100** | 87.5 | 78.5 |
| HPE@Lavie | **80** | **80** | **80** | **100** | **100** | 75 | **100** | **87.9** |

| AD@PLCC(↑) | AE | IA | PV | TJ | KE | SF | Overall |
|---|---|---|---|---|---|---|---|
| GPT-4o | 0.37 | 0.22 | 0.23 | 0.28 | 0.37 | 0.18 | 0.28 |
| HPE | **0.71** | **0.57** | **0.50** | **0.58** | **0.65** | **0.58** | **0.60** |
| GPT-4o@OpenSora | 0.22 | -0.39 | 0.32 | **0.15** | -0.03 | -0.12 | 0.03 |
| HPE@OpenSora | **0.37** | **0.55** | **0.34** | 0.06 | **0.28** | **0.41** | **0.34** |
| GPT-4o@Lavie | 0.17 | 0.13 | -0.34 | 0.06 | -0.09 | -0.15 | -0.04 |
| HPE@Lavie | **0.28** | **1.0** | **0.49** | **0.37** | **0.12** | **0.69** | **0.49** |

| RM@PLCC(↑) | AE | BC | FC | IA | PV | TJ | EI | Overall |
|---|---|---|---|---|---|---|---|---|
| GPT-4o | 0.07 | 0.18 | 0.20 | 0.32 | -0.14 | -0.01 | -0.14 | 0.07 |
| HPE | **0.52** | **0.43** | **0.43** | **0.43** | **0.20** | **0.56** | **0.44** | **0.43** |
| GPT-4o@OpenSora | -0.45 | -0.03 | **0.08** | 0.0 | 0.04 | -0.23 | 0.14 | -0.06 |
| HPE@OpenSora | **0.25** | **0.35** | 0.05 | **0.42** | **0.89** | **0.89** | **0.44** | **0.47** |
| GPT-4o@Lavie | 0.11 | -0.07 | 0.42 | **0.42** | 0.21 | 0.31 | -0.21 | 0.17 |
| HPE@Lavie | **0.33** | **0.04** | **0.69** | 0.40 | **0.89** | **0.67** | **0.06** | **0.44** |

Table 6: **Performance comparison between Human Preference Evaluatorand GPT-4o.** HPE indicates Human Preference Evaluator. The other abbreviations are listed in Tab. 2.

In Autonomous Driving, we fine-tune using the nuScenes training set (Caesar et al., 2020), and following the approach in Vista (Gao et al., 2024), we sample video clips consisting of 25 frames at a frequency of 10 Hz. To classify actions into textual commands, we adhere to established conventions in planning and define ego-vehicle commands as "turn right", "turn left", "go straight", and "stop", consistent with the definitions in Vista.

In Robot Manipulation, we use RH20T-P (Chen et al., 2024), a dataset based on RH20T (Fang et al., 2023) and designed for primitive-level robotic manipulation that features meticulously defined primitive skills and diverse primitive-level spatial knowledge of multiple forms. We use each primitive-level robotic manipulation instruction along with the corresponding video as input for training. Additionally, since this dataset is designed for downstream tasks in specific scenarios, some textual instructions include explicit coordinate information. To enhance the generalization ability of the video model, we excluded these coordinate-specific instructions during training.

At the model architecture level, we followed Dynamicrafter (Xing et al., 2023) to modify the text-to-video model of Open-Sora-Plan(T2V) (Lab & etc., 2024) by replacing the first frame and expanding the channel dimensions, enabling the model to take the first frame as a condition. This resulted in the Open-Sora-Plan (TI2V) model. No structural adjustments were made to other models. During training, we preprocessed the data according to each model's default input format and performed fine-tuning following the official implementation without changing the training settings. We fine-tuned each model using two different video lengths to enhance the diversity of the video evaluation set: short videos with approximately 20 frames and long videos with around 60 frames, depending on the model's default training video length. The specific lengths are detailed in the Tab. 5.

### B.3 HUMAN PREFERENCE EVALUATOR TRAING

The Human Preference Evaluator is trained based on Flash-VStream (Zhang et al., 2024a), where only LoRA (Hu et al., 2021) parameters are trained. The model's input consists of a sampled video, represented as multiple frames, along with a prompt. The prompt includes the current scenario, the instruction input for video generation, the dimension being evaluated, and the definition of that

> <Video>\nThe given autonomous driving video is generated by a generative model based on the input instruction: {instruction}. Please rate the video based on the following criteria: {Dimension}: {Dimension Explanation}

Figure 6: **Prompt template for Autonomous Driving**. The {item} is replaced with specific content.

Table 7: **Evaluation results in OE**. The abbreviations are listed in Tab. 2.

| Model | BC | FC | IA | SA | VC | TJ | EI | Overall |
|---|---|---|---|---|---|---|---|---|
| Open-Sora-Plan | 1.4 | 1.9 | 1.7 | 1.7 | **2.0** | 1.5 | 1.6 | 1.69 |
| Lavie | 1.3 | **2.0** | 1.7 | 1.7 | **2.0** | **2.0** | **1.8** | 1.79 |
| ModelScope | **1.9** | **2.0** | **2.0** | 1.7 | **2.0** | **2.0** | 1.75 | **1.91** |
| OpenSora | 1.6 | 1.9 | 1.6 | 1.8 | **2.0** | **2.0** | 1.6 | 1.79 |
| AnimateDiff | 1.3 | 1.3 | 1.2 | 1.7 | 1.4 | 1.38 | 1.55 | 1.40 |
| DynamicCrafter | **1.9** | **2.0** | 1.5 | **2.0** | **2.0** | **2.0** | 1.45 | 1.84 |
| EasyAnimate | 1.4 | 1.8 | 1.5 | **2.0** | **2.0** | 1.22 | 1.45 | 1.62 |

dimension. An example of such a prompt is illustrated in Fig. 6, while the details of the explanation are discussed in Section 2. We don't use the annotated reason during training for CoT of the evaluator, as the reason labeled by different human varies a lot, hard for model to learn.

We maintain consistent training settings in all three scenarios, with a video sampling frequency of 4. The LoRA settings aligned with those in Flash-VStream. We use AdamW as the optimizer, employ cosine decay for the learning rate scheduler. We train for 4 epochs with a learning rate of 2e-5 and a warmup ratio of 0.03. The training is conducted on 4 A100 80 GPUs. To avoid over-fitting to specific prompts or videos generated by particular models, we carefully filter the HF-Embodied Dataset to ensure balanced distribution across various generation models and evaluation dimensions.

We prove the effectiveness and generalizability of through comparison with GPT-4o arcoss the three embodied scenarios, under both finetuned and zero-shot setting, as shown in Tab. 6. After fine-tuning, the Human Preference Evaluator surpasses GPT4-o in aligning with human preferences across all dimensions in every scenario. This is particularly evident in challenging dimensions, *e.g.*, Embodied Interaction and Trajectory in RM, where GPT4-o shows a negative correlation, while the Human Preference Evaluator exhibits a strong positive correlation. These results demonstrate the its robust performance, making it suitable for Explicit Perceptual Evaluation. In zero-shot settings, the Human Preference Evaluator also outperforms GPT4-o in nearly all dimensions, further proving our model's aility to understand videos generated by different models.

## C    DETAILED RESULT OF EXPLICIT PERCEPTUAL EVALUATION

### C.1    QUANTITATIVE RESULTS

Tabs. 7-9 present the comprehensive evaluation results for 7 video generation models across three scenarios, including the scores for each dimension and the mean scores representing the overall performance of the models. In OE, although our scoring is binary, we display scores on a scale of 1-2 for consistent comparison. In addition to the conclusions mentioned in the main text, we can observe the following findings.

In OE, most models achieve high scores in Velocity, largely due to the limited occurrences of object movement in the generated videos. Generating dynamic embodied environments with moving objects presents a significant challenge for current models. Additionally, the consistency between the generated videos and the scenarios specified in the instructions is higher than the alignment with the task-oriented instructions. This indicates that while the models can generate corresponding scenes, they struggle to reason about the temporal actions necessary for task completion.

In AD, the quality of the generated videos significantly declines due to the complexity of outdoor driving scenarios. The models must understand and generate various traffic elements, *e.g.*, roads,

Table 8: **Evaluation results in AD.** The abbreviations are listed in Tab. 2.

| Model | AE | IA | PV | TJ | KE | SF | Overall |
|---|---|---|---|---|---|---|---|
| Open-Sora-Plan | 1.6 | **5.0** | 1.55 | 1.4 | 1.45 | 3.2 | 2.37 |
| Lavie | 2.15 | **5.0** | 2.2 | 2.8 | 2.1 | **5.0** | 3.21 |
| ModelScope | 2.8 | **5.0** | 3.35 | 4.0 | 3.0 | **5.0** | 3.86 |
| OpenSora | **3.55** | **5.0** | **4.4** | **4.8** | **3.65** | **5.0** | **4.40** |
| AnimateDiff | 1.55 | **5.0** | 1.55 | 1.0 | 1.3 | 3.8 | 2.37 |
| DynamicCrafter | 2.6 | 4.0 | 3.4 | 3.8 | 2.65 | 5.0 | 3.57 |
| EasyAnimate | 1.5 | 3.4 | 1.4 | 1.4 | 1.3 | 2.6 | 1.93 |

Table 9: **Evaluation results in RM.** The abbreviations are listed in Tab. 2.

| Model | AE | BC | FC | IA | PV | TJ | EI | Overall |
|---|---|---|---|---|---|---|---|---|
| Open-Sora-Plan | **4.0** | 4.0 | **4.0** | 1.0 | 4.9 | **5.0** | 4.0 | 3.84 |
| Lavie | 3.8 | 3.9 | **4.0** | 1.8 | 4.95 | **5.0** | 4.1 | 3.94 |
| ModelScope | 3.63 | 4.1 | **4.0** | 1.18 | 4.9 | **5.0** | 4.0 | 3.83 |
| OpenSora | 3.85 | 4.0 | 3.95 | 1.3 | 4.75 | **5.0** | 4.1 | 3.85 |
| AnimateDiff | 3.8 | 3.9 | **4.0** | 1.0 | 4.95 | **5.0** | 4.1 | 3.82 |
| DynamicCrafter | 3.97 | **4.08** | **4.0** | **2.6** | **5.0** | **5.0** | **4.31** | 4.14 |
| EasyAnimate | 3.55 | 3.45 | 3.65 | 1.2 | 4.8 | 4.3 | 3.45 | 3.49 |

background buildings, pedestrians, and vehicles, while also producing dynamic content, with each element requiring reasonable speed. This presents substantial challenges. However, top-performing models, *e.g.*, OpenSora, manage to achieve the highest scores across all metrics.

In RM, the primary issue lies in Instruction Alignment. The video generation models struggle to comprehend the input instructions and generate appropriate actions to complete the tasks, instead moving aimlessly without clear objectives. This lack of targeted movement reduces potential errors related to object interaction or penetration, resulting in artificially inflated scores in Embodied Interaction and Trajectory. Current video generation models struggle in effectively addressing robotic manipulation tasks.

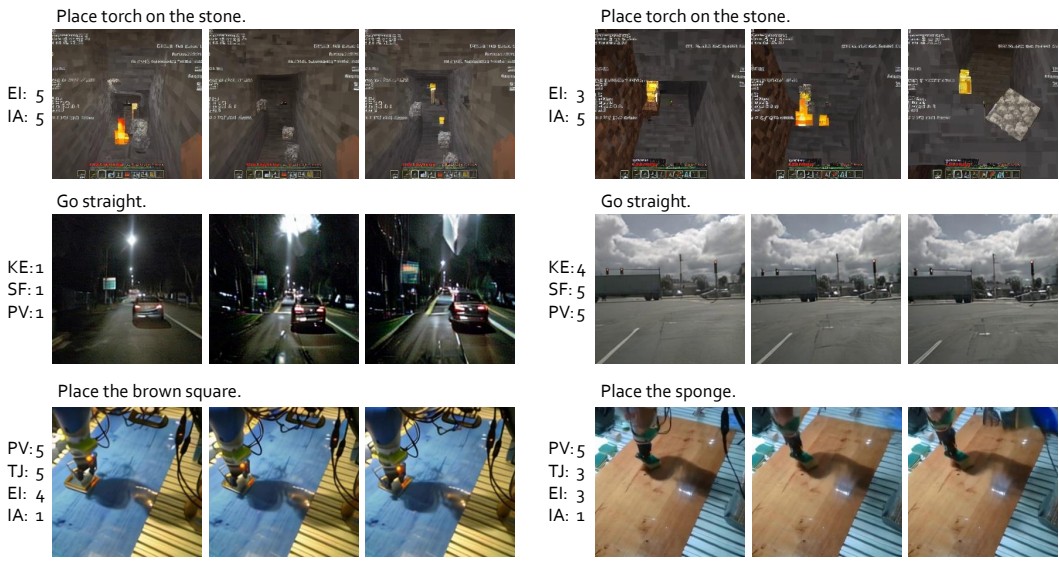

Figure 7: Qualitative Results in Explicit Perceptual Evaluation.

## C.2 QUALITATIVE RESULTS

We include a qualitative analysis of generated videos under the three embodied scenarios. Each video is represented by three evenly sampled frames, with the corresponding generation instructions listed above the video. To the left of the videos, we provide the scores of the key embodied attributes labeled by the human preference evaluator.

**Open-Ended Embodiment Scenario.** For the open-ended embodiment scenario, the left video demonstrates successful completion of the instructed task, with proper interaction with the stone. In contrast, the right video encounters issues during interaction, specifically crushing the stone when placing the torch, indicating a problematic interaction.

**Autonomous Driving Scenario.** In the autonomous driving context, the left video suffers from significant distortion and light pollution. Additionally, it exhibits unsafe behavior, such as maintaining excessive speed despite the presence of a car ahead. On the other hand, the right video maintains high-quality generation and demonstrates proper adherence to traffic rules, including slowing down at a red light.

**Robotic Manipulation Scenario.** For robotic manipulation, the left video displays that the robotic arm interacts with a rigid object (wooden block), which appropriately does not deform during the grasping process. However, minor, physically implausible rotations occur during the grasp, resulting in a score of 4 for EI. Additionally, the generated wooden block does not match the specified color in the instruction, leading to an instruction alignment score of 1. In contrast, the flexible object (sponge) in the right video, unrealistically stretched, violating physical rules. Furthermore, the video depicts the robotic arm moving away from the table, which contradicts the "place" instruction. This mismatch leads to low scores in both trajectory and instruction alignment. Despite these issues, both videos effectively display light reflections and shadows, with a clear sense of depth, earning a PV score of 5.

These qualitative results provide an illustration of "what is a good embodied video", and reveal the limitations of the video generation models.

## D  IMPLICIT MANIPULATIVE EVALUATION-OE

In this section, we provide additional details about Implicit Manipulative Evaluation-Open-Ended Embodied Environment that are not covered in the main paper due to space limitations. Minecraft has emerged as a popular open-world environment for developing generalist embodied agents (Lifshitz et al., 2024; Qin et al., 2024; Zhou et al., 2024) due to its diverse tasks (e.g., survival, harvesting, crafting, combat, and creative tasks), varied environments, and interactive mobs, all of which require generalized agent capabilities. Previous works (Qin et al., 2024; Wang et al., 2023d;a) have primarily focused on exploring the capabilities of LLMs or MLLMs as Predictive Text Modelat the $S_1$ stage. However, no prior research has conducted closed-loop evaluations of World Simulators at the $S_3$ stage within Minecraft. To address this gap, we leverage the Steve-1 pipeline to assess the performance of Video Generation Models as World Simulators in Open-Ended Embodied Environment.

### D.1  DETAILED DESCRIPTION

In Implicit Manipulative Evaluation-Open-Ended Embodied Environment, we adapt the action space of Steve-1 (Lifshitz et al., 2024) to develop a pipeline for the Video Generation Model, enabling it to function as a low-level embodied controller. Additionally, we employ Programmatic Evaluation to benchmark the low-level embodied control capabilities of the Video Generation Model as World Simulators. These tasks are comprehensive, requiring the combination of multiple atomic actions and smooth scene transitions. Each aspect rigorously tests the coherence of the generated content, the consistency with given instructions, and the model's ability to interact effectively with the environment.

**Testing.** We evaluated performance in OE using five tasks: collecting wood, collecting dirt, collecting seeds, exploring the area, and vertical digging. To reduce evaluation randomness, we selected the most suitable initialization environments for each task (e.g., the agent is initialized in a forest for the wood collection task). During testing, for each task, we randomly select one description

from various task instructions and input it into the World Simulator to generate the corresponding video. The video is then continuously translated into actions by a pre-trained goal-based video-to-action model, which executes until the test time expires. Each task runs for 10 trials with distinct environment seeds, with a limit of 3,000 frames (*i.e.*, 2.5 minutes of gameplay).

**Training.** Due to the low video quality produced by the open-source video generation model based on the provided instructions, we applied additional fine-tuning using data from the OE simulation environment. For Video Generation Model fine-tuning, we use OpenAI Contractor Gameplay Dataset (Baker et al., 2022) which is the same as OE in Explicit Perceptual Evaluation. The training setting could be found in Sup. B.2. For pre-trained goal-based video-to-action model, we use pre-trained Steve-1(visual) model without extra fine-tuning.

**Metrics.** We calculate programmatic evaluation metrics by tracking the MineRL environment state throughout each evaluation episode. Several metrics are measured, including travel distance and early-game item collection. Travel distance is defined as the agent's maximum displacement on the horizontal (X-Z) plane from its initial spawn point. Dig depth is defined as the agent's maximum displacement on the vertical (Y) axis from its initial spawn point. For an early-game inventory, we record the maximum count of logs, seeds, and dirt items observed in the agent's inventory during the episode.

### D.2 ACTIONS

We use the part of the action space of (Baker et al., 2022) which encompasses nearly all actions available to human players, including keypresses, mouse movements, and clicks. The specific binary actions used in our setup are listed in Tab 10.

Table 10: **Action Space of OE.**

| Behavior | Action |
|---|---|
| forward | W key |
| back | S key |
| left | A key |
| right | D key |
| jump | space key |
| inventory | E key |
| sneak | shift key |
| sprint | ctrl key |
| attack | left mouse button |

### D.3 FULL RESULT

Tab. 11 presents the evaluation results of several models across five specific tasks (collect wood, collect dirt, collect seeds, travel distance, and dig depth), along with the average (AVG) score for each model. The models are evaluated under two different conditions: Text and Text & Image. Notably, to ensure that each task falls within a similar score range, we divided the score for the travel distance task by 10 to calculate the AVG score.

**Performance of Models Under Text Condition.** Open-Sora-Plan and Lavie demonstrate strong performance under the text-only condition, especially in the collect dirt and travel distance tasks. Their average scores (26.38 and 26.06, respectively) are very close, indicating consistent and robust performance across tasks. ModelScope shows an excellent score in the collect dirt task (52.20), but it performs poorly in tasks like collect wood (14.00) and travel distance (240.72), resulting in an overall lower average score (21.050) compared to other text-based models. OpenSora stands out with the highest overall average score (27.80), excelling particularly in collect dirt (70.20) and travel distance (339.87). This suggests that it is well-adapted to a variety of tasks and exhibits strong task performance. AnimateDiff shows the weakest performance across all tasks, especially in collect wood (7.40) and collect seeds (3.30), indicating challenges in handling such tasks.

**Performance of Models Under Text & Image Condition.** Open-Sora-Plan shows a significant drop in average score under the "Text & Image" condition, demonstrating that adding image input reduces its performance compared to the text-only condition. In particular, its travel distance score drops from 342.91 to 195.14, suggesting that incorporating image data might interfere with certain tasks. DynamICrafter and EasyAnimate exhibit poor performance across all tasks, especially in collect wood and collect seeds, where they barely complete the tasks (with scores of 0.40 and 0.20, respectively). This may indicate a lack of generalization ability in these models when combining image input with text. Comparing the "Text" and "Text & Image" conditions, we observe that adding image input does not consistently improve task performance and, in some cases, even degrades it. We also observed that the success rates of various tasks significantly decrease when an image is added as an additional condition. This indicates that the current video generation models need improvement in handling multiple conditional inputs.

Table 11: Detail Result of Open-Ended Embodied Environment in Implicit Manipulative Evaluation.

| Model | Condition | AVG | Specific Tasks | | | | |
| --- | --- | --- | --- | --- | --- | --- | --- |
| | | | Collect Wood | Collect Dirt | Collect Seed | Travel Dis. | Dig Depth |
| Open-Sora-Plan | | 26.38 | 19.90 | 50.20 | 7.30 | 342.91 | 20.20 |
| Lavie | | 26.06 | 23.50 | 56.00 | 11.60 | 270.20 | 12.20 |
| ModelScope | Text | 21.050 | 14.00 | 52.20 | 6.30 | 240.72 | 8.70 |
| OpenSora | | 27.80 | 21.20 | 70.20 | 10.40 | 339.87 | 3.20 |
| AnimateDiff | | 13.10 | 7.40 | 22.90 | 3.30 | 274.19 | 4.50 |
| Open-Sora-Plan | | 10.28 | 11.10 | 12.50 | 2.60 | 195.14 | 5.70 |
| DynamiCrafter | Text & Image | 4.06 | 0.40 | 0.30 | 1.30 | 130.04 | 5.30 |
| EasyAnimate | | 4.84 | 0.20 | 0.70 | 1.70 | 157.12 | 5.90 |

## D.4 ROLL OUT

Fig. 8 illustrates the downstream execution process in the Open-Ended Embodied Environment, along with the corresponding textual instructions.

Insturction: Collect wood in the forest.

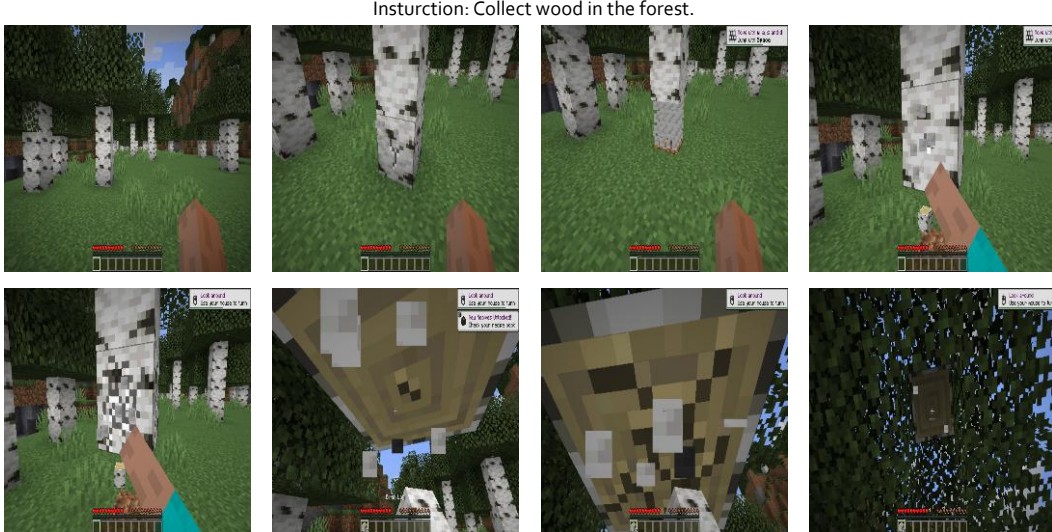

Figure 8: Rollout of Open-Ended Embodied Environment in Implicit Manipulative Evaluation.

## E IMPLICIT MANIPULATIVE EVALUATION-AD

In this section, we provide additional details about Implicit Manipulative Evaluation-Autonomous Driving that are not covered in the main paper due to space limitations.

### E.1 DETAILED DESCRIPTION

In Implicit Manipulative Evaluation-Autonomous Driving, we adapt the action space of LM-Drive (Shao et al., 2024) to develop a pipeline for the Video Generation Model, enabling it to function as a low-level embodied controller. Additionally, we employ LangAuto (Language-guided Autonomous Driving) CARLA benchmark, to evaluate the low-level embodied control capabilities of the Video Generation Model as World Simulators. These tasks are designed to be comprehensive, spanning all 8 publicly available towns in CARLA, covering a diverse range of scenarios *e.g.*, highways, intersections, and roundabouts. Additionally, they account for 16 different environmental conditions, combining 7 distinct weather settings (Clear, Cloudy, Wet, MidRain, WetCloudy, HardRain, SoftRain) with 3 daylight conditions (Night, Noon, Sunset). Each aspect rigorously tests the coherence of the generated content, the consistency with given instructions, and the model's ability to interact effectively with the environment.

**Testing.** We evaluated performance in Autonomous Driving using the LangAuto-Tiny benchmark setting where the route length is shorter than 150 meters. We posit that shorter driving distances provide a more effective test of the low-level control capabilities of World Simulators. Longer routes typically involve more instructions, which are prone to misalignment with the real-time simulation environment. Therefore, we opt to evaluate performance on shorter routes to minimize these discrepancies. During testing, we randomly select one description from various task instructions and input it into the World Simulator to generate the corresponding video. The video is then continuously translated into actions by a pre-trained goal-based video-to-action model, which executes until the test time expires. We use the corresponding LangAuto-Tiny instructions and the first-person view rendered by the real-time CARLA simulation environment as input to the video generation model. The generated video is then continuously transformed into downstream control signals using a pre-trained video-to-action model until the agent reaches a predefined success zone or the task is terminated due to factors *e.g.*, timeouts or collisions.

**Training.** Due to the low video quality produced by the open-source video generation model based on the provided instructions, we applied additional fine-tuning using data from the AD simulation environment. For Video Generation Model training, we use LMDrive Training Dataset (Shao et al., 2024). We preprocessed the training data according to each model's default input format and performed fine-tuning following the official implementation without changing the training settings. We fine-tuned each model using a short video generation setting with approximately 20 frames. For the video-to-action model, we use pre-trained LMdrive model. Additional fine-tuning was conducted based on the test requirements. We provided the model with arbitrary text instructions and replaced the visual input with the future frame while keeping all other training settings consistent with LM-Drive.

**Metrics.** We consider eight key metrics introduced by the CARLA Leaderboard (Dosovitskiy et al., 2017): Route Completion (RC), Infraction Score (IS), Driving Score (DS), Vehicle Collisions (VC), Pedestrian Collisions (PC), Layout Collisions (LC), Red Light Violations (RV), and Offroad Infractions (OI). Route Completion refers to the percentage of the total route length that the agent has completed. This metric only accounts for the distance traveled along the predetermined route, where each segment corresponds to a navigation instruction. If the agent strays too far from the route, it is considered to have violated the instruction, resulting in the episode being marked as a failure and terminated. The Infraction Score tracks any infractions caused by the agent, with penalties applied for collisions or traffic violations through a corresponding discount factor. The Driving Score is the product of the route completion ratio and the infraction score, reflecting both driving progress and safety, and is widely regarded as the primary ranking metric. The precise definitions of the residual metrics can be found in the CARLA documentation (Dosovitskiy et al., 2017).

### E.2 ACTIONS

The video generated by the World Simulator is continuously fed into the video-to-action model to obtain the corresponding waypoints. The agent then generates control signals based on the generated waypoints and the conversion strategy used in CARLA.

### E.3 FULL RESULT

Tab. 12 presents the evaluation results of several models across eight metrics. The evaluation results highlight significant differences in how video generation models perform in autonomous driving tasks. Open-Sora-Plan stands out in trajectory generation, instruction following, and environment perception, producing high-quality videos that effectively support task execution. In contrast, DynamiCrafter and EasyAnimate struggle with generating detailed and consistent video content, particularly when handling complex or dynamic scenes. These models require improvements in video generation quality, scene understanding, and task alignment to enhance their performance.

From a video generation perspective, several key areas for future development are identified: Improved Trajectory Generation: High-quality trajectory generation is essential for accurate control signals. Models must focus on generating more coherent and precise trajectories, especially in dynamic environments, to ensure vehicles follow instructions and avoid collisions. Enhanced Instruction Following: Generated videos should closely align with task instructions, particularly in changing environments, enabling vehicles to adapt quickly while maintaining task accuracy. Better Environment Perception: Future models need to generate videos that accurately represent complex scenes, *e.g.*, interactions with pedestrians, other vehicles, and varied terrains. More detailed and realistic video generation will provide stronger input for real-time decision-making in the control system.

In summary, advancing trajectory accuracy, instruction alignment, and environment representation will be crucial for improving the overall performance of these video generation models in autonomous driving tasks.

Table 12: Detail Result of Autonomous Driving in Implicit Manipulative Evaluation.

| Model | DS($\uparrow$) | RC($\uparrow$) | IS($\uparrow$) | VC($\downarrow$) | PC($\downarrow$) | LC($\downarrow$) | RV($\downarrow$) | OI($\downarrow$) |
|---|---|---|---|---|---|---|---|---|
| Open-Sora-Plan | 31.054 | 38.249 | 0.767 | 2.400 | 0.000 | 4.401 | 1.133 | 3.514 |
| DynamiCrafter | 24.491 | 37.189 | 0.599 | 5.030 | 0.000 | 4.896 | 0.937 | 3.221 |
| EasyAnimate | 17.414 | 28.475 | 0.607 | 0.000 | 0.000 | 29.344 | 0.000 | 1.690 |

### E.4 ROLL OUT

Fig. 9 illustrates the downstream execution process in the Autonomous Driving, the corresponding text instructions can be found in the lower left corner of each frame.

## F IMPLICIT MANIPULATIVE EVALUATION-RM

In this section, we provide additional details about Implicit Manipulative Evaluation-Robot Manipulation that are not covered in the main paper due to space limitations.

### F.1 DETAILED DESCRIPTION

We primarily conduct our experiments on the CALVIN benchmark (Mees et al., 2022), which is specifically designed for long-horizon, language-conditioned manipulation tasks. CALVIN includes four simulated environments (labeled A, B, C, and D) that differ in textures and object placements. Each environment features a Franka Emika Panda robot positioned next to a desk with various manipulable objects. The evaluation protocol tests model performance across 1,000 unique instruction chains, each consisting of five distinct tasks. By providing an extensive dataset paired with natural language annotations, the CALVIN benchmark can provide a close-loop evaluation platform for evaluating World Simulator to test its generation and generalization capabilities.

**Testing.** We evaluated performance in Robot Manipulation using the CALVIN benchmark benchmark, policy models are trained on demonstrations from environments A, B, and C, and evaluated in a zero-shot manner in environment D. During the testing phase, we leverage World Simulators and a pre-trained video-to-action model to tackle novel manipulation tasks guided by user-specified natural language commands. Given a current observation, we generate future video predictions using

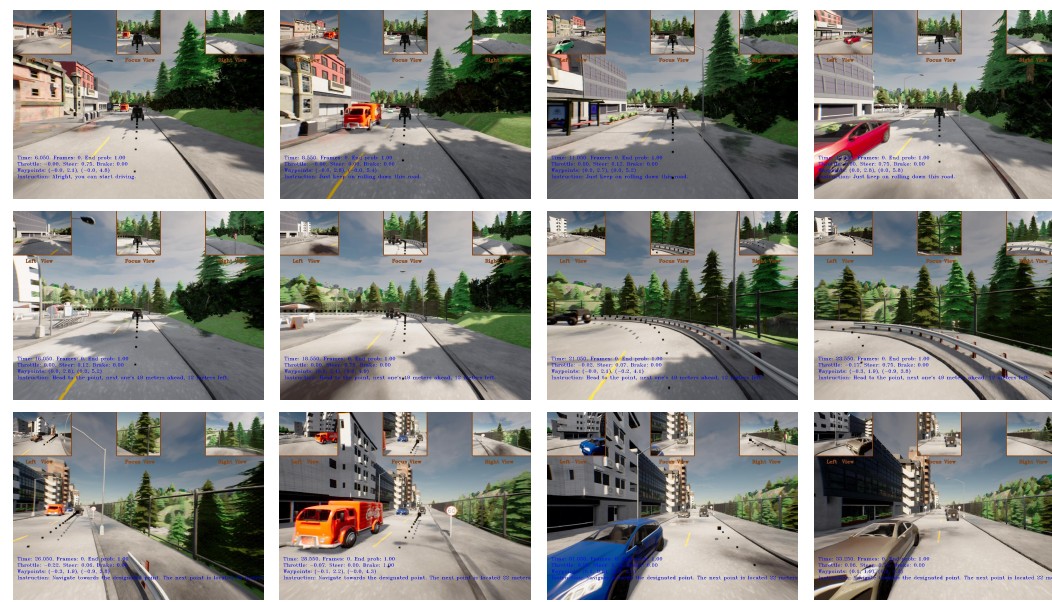

Figure 9: Rollout of Autonomous Driving in Implicit Manipulative Evaluation.

the World Simulator for the manipulation task with text instruction. Once the video is sampled, we then execute the video-to-action policy conditioned on for $k$ timesteps, where $k$ is a testing hyperparameter. After $k$ timesteps, the video prediction is refreshed by sampling from the World Simulator again, and the process is repeated.

**Training.** Due to the low video quality produced by the open-source video generation model based on the provided instructions, we applied additional fine-tuning using data from the RM simulation environment. For Video Generation Model training, we use Calvin(ABC_D) datset (Mees et al., 2022). We preprocessed the training data according to each model's default input format and performed fine-tuning following the official implementation without changing the training settings. We fine-tuned each model using a short video generation setting with approximately 20 frames. For the video-to-action model, we use a pre-trained Susie policy without extra fine-tuning.

**Metrics.** We report the success rates and the average task length completed (out of five tasks) for each evaluation sequence.

### F.2 ACTIONS

For low-level control, we utilize the same action space as Calvin (Mees et al., 2022).

### F.3 FULL RESULT

Based on the results shown in Tab. 13, Open-Sora-Plan demonstrates consistent performance, with an average task length of 2.95, indicating its ability to reliably complete task sequences. While DynamiCrafter achieves a higher success rate of 0.95 on the initial task, its performance declines as task complexity increases, suggesting limitations in handling longer manipulation sequences. EasyAnimate, although moderately successful in completing early tasks, experiences a sharp decline in performance as task difficulty rises, reflected in its lower average task length of 2.05.

Overall, the models' ability to consistently complete multiple tasks in succession showcases their potential in downstream applications, with Open-Sora-Plan emerging as the most capable. However, the observed decrease in success rates as task complexity increases highlights the need for further improvements in video-to-action translation, particularly in addressing the challenges posed by longer and more complex manipulation sequences.

Table 13: Detail Result of Robot Manipulation in Implicit Manipulative Evaluation.

| Method | Task completed in a row (%) ↑ | | | | | Avg. Len. ↑ |
|---|---|---|---|---|---|---|
| | 1 | 2 | 3 | 4 | 5 | |
| Open-Sora-Plan | 0.85 | 0.70 | 0.60 | 0.40 | 0.40 | 2.95 |
| DynamiCrafter | 0.95 | 0.75 | 0.55 | 0.25 | 0.25 | 2.75 |
| EasyAnimate | 0.90 | 0.60 | 0.35 | 0.10 | 0.10 | 2.05 |

To minimize the impact of randomness caused by the number of experiments, we conducted an additional 100 trajectories evaluation. The results are presented in Tab 14. Compared to the 20-trajectories setup, the results from the 100-trajectories setup show slight variations but maintain a consistent overall trend. We also compared the performance with Unipi based on the 25-trajectory setup described in SuSIE, and it can be observed that the tested video generation models(Open-Sora-Plan, DynamiCrafter, EasyAnimate) demonstrate superior capabilities compared to the PVDM Yu et al. (2023) latent video diffusion model utilized by Unipi.

Table 14: Detail Result of Robot Manipulation in Implicit Manipulative Evaluation, by running 100 trajectories. ∗ Results reported by Susie Black et al. (2023).

| Method | Task completed in a row (%) ↑ | | | | | Avg. Len. ↑ |
|---|---|---|---|---|---|---|
| | 1 | 2 | 3 | 4 | 5 | |
| UniPi∗(HiP) | 0.08 | 0.04 | 0.00 | 0.00 | 0.00 | - |
| UniPi∗ (Susie) | 0.56 | 0.16 | 0.08 | 0.08 | 0.04 | - |
| Open-Sora-Plan | 0.89 | 0.72 | 0.63 | 0.34 | 0.32 | 3.12 |
| DynamiCrafter | 0.93 | 0.69 | 0.51 | 0.27 | 0.18 | 2.64 |
| EasyAnimate | 0.92 | 0.55 | 0.32 | 0.16 | 0.13 | 2.08 |

## F.4 ROLL OUT

Fig. 10 illustrates the downstream execution process in the Robot Manipulation, along with the corresponding textual instructions.

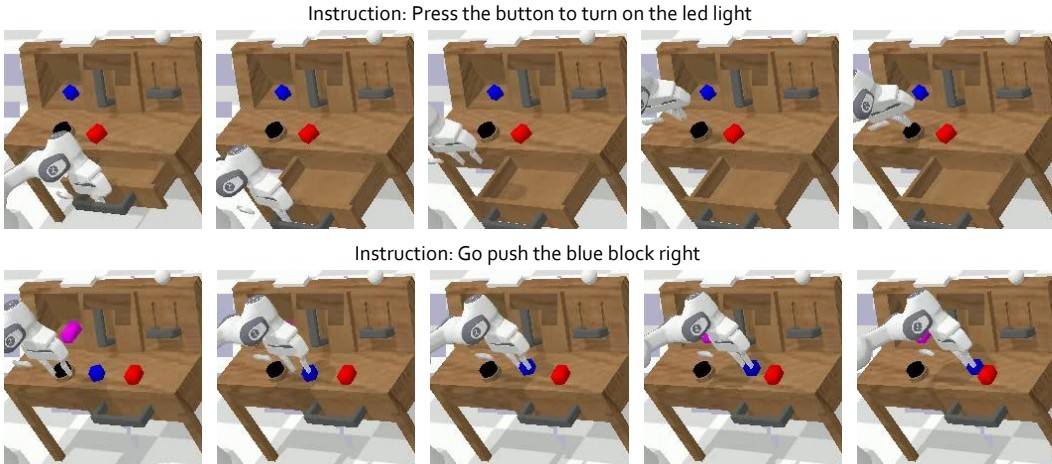

Figure 10: Rollout of Robot Manipulation in Implicit Manipulative Evaluation.

## G DISCUSSION OF VISION-LANGUAGE-ACTION MODELS

The generated videos have the potential to significantly enhance the performance of Vision-Language-Action (VLA) models by addressing two key challenges in training such models: the

availability of diverse, high-quality training data and the need for effective reward functions in real-world scenarios.

**Data Augmentation and Hindsight Relabeling for Imitation Learning.** Generated videos can serve as a valuable source of synthetic data for training VLA models. By leveraging the diversity and scalability of generative models, we can create a wide array of training scenarios, covering edge cases and rare events that are difficult to capture in real-world datasets. Additionally, these videos enable hindsight relabeling, a process where we retrospectively adjust the labels of generated data to align with desired outcomes. This approach is particularly effective for imitation learning, allowing VLA models to learn optimal behavior by mimicking successful trajectories represented in the generated videos. By expanding the data distribution and improving its quality, generative videos can lead to more robust and generalizable VLA models.

**Reward Generation for Online Reinforcement Learning.** Beyond data augmentation, generated videos can act as a Reward Generator in reinforcement learning (RL) contexts. Unlike traditional RL setups that rely on pre-defined reward functions within a simulator, generative videos enable the creation of dense and context-aware reward signals tailored to real-world tasks. For example, they can simulate desirable outcomes or intermediate goals, providing detailed feedback to the agent. This capability is particularly crucial for transferring RL models to real-world environments, where designing explicit reward functions is often impractical. By aligning the generated rewards with real-world objectives, we can bridge the gap between simulation and reality, allowing VLA models to achieve higher performance in real-world tasks.

