# OpenReview forum: "WorldSimBench: Towards Video Generation  Models as World Simulators"
_ICLR.cc/2025/Conference — Submitted to ICLR 2025_

### Official Review · Reviewer_kNpR · 2024-11-02

**Soundness:** 2
**Presentation:** 3
**Contribution:** 2
**Rating:** 5
**Confidence:** 5

**Summary:**

This paper introduced WorldSimBench, a dual evaluation framework for assessing World Simulators by conducting thorough evaluations through Explicit Perceptual Evaluation and Implicit Manipulative Evaluation, and examined several video generation models as World Simulators in several simulators.

**Strengths:**

1. This paper provides a solid overview of the world simulator.
2. The dataset construction involves the introduction of costly human preference annotations, which adds meaningful data to the study.
3. The experimental design is quite comprehensive, covering a wide range of aspects.

**Weaknesses:**

1. First, this article integrates video generation with downstream tasks, but this kind of idea has been around for a long time. For example, this is not the first paper to apply video generation to robotic tasks; the first one should be UniPi. Additionally, the authors did not innovate on the paradigm of combining with downstream tasks, and the overall approach still references the video-action mapping method from SuSIE. Furthermore, even following SuSIE's methodology, it should not be applied to video generation since SuSIE is a framework for goal image generation. The authors should have used the UniPi framework here, although this would lead to a drop in results.

2. There are issues with the evaluation metrics. For example, in the CALVIN task, the common practice is to test using an evaluation method of 1000/100 (1000 times for standard evaluation and 100 times for the UniPi method), which ensures coverage of all 34 task categories. However, the authors only used 20 evaluations, resulting in a significant number of generated tasks not being tested, leading to a lack of strong credibility in the current results. Moreover, there is no comparison with SOTA methods.

3. The most valuable aspect of the entire article is the generation of human feedback annotations. However, the authors did not leverage this feedback for further optimization, merely stopping at obtaining a score. If they could use the human feedback annotation data to enhance alignment in downstream tasks through RLHF, the innovation of this article would be more substantial.

4. In the methods section of Figure 3, there is a gap between the authors' testing of RM performance. The authors still used the original annotations from the dataset and did not apply a high-level planner to break down tasks. This results in a gap between the framework and experimental parts. Such an organizational structure exaggerates the generality of the method, but when solving practical problems, traditional methods are still employed. This indicates that the framework's design is not advancing the task further; it merely replaces it with a better video generation model, which still stems from the performance improvements of Open-Sora.

**Questions:**

I understand that the authors intend to establish a comprehensive new paradigm for video generation that can address many issues, especially the downstream video-to-action problem. However, the article should focus more on how to better utilize human feedback annotations to improve the downstream video-to-action problem. Without discussing this point, most of the content mentioned in the article is already existing and established, resulting in a lack of substantial innovation.

---

> ### Author Response · Authors · 2024-11-21
> **Response to kNpR (1/3)**
>
> Thank you for your constructive and thoughtful comments. They were indeed helpful in improving the paper. We take this opportunity to address your concerns:
>
> ### List of changes in the manuscript:
> 1. Section B.1 in Appendix. Discussions about the future work of HF-Embodied dataset.
> 2. Section F.3 in Appendix. Evaluation with 100-trajectories and compared with Unipi.
>
> ---
>
> ### Question: Misunderstanding of the motivation and contribution of the paper.
>
> **A:**  We appreciate your valuable feedback and would like to address the misunderstanding regarding our paper’s motivation and contributions, which appears to have influenced your assessment.
>
> **Clarification of Motivation and Contribution**
> Your comment, *"I understand that the authors intend to establish a comprehensive new paradigm for video generation that can address many issues, especially the downstream video-to-action problem,"* does not accurately reflect our intentions.
>
> Our primary objective is not to propose a method for improving downstream tasks. Instead, as stated at the beginning of the abstract, *"Recent advancements in predictive models have demonstrated exceptional capabilities in predicting the future state of objects and scenes. However, the lack of categorization based on inherent characteristics continues to hinder the progress of predictive model development. Additionally, existing benchmarks are unable to effectively evaluate higher-capability, highly embodied predictive models from an embodied perspective. In this work, we classify the functionalities of predictive models into a hierarchy and take the first step in evaluating World Simulators by proposing a dual evaluation framework called WorldSimBench."*
>
> Our work focuses on evaluating and analyzing the capability of video generation models as World Simulators, addressing the limitations of traditional video evaluation methods in assessing attributes unique to world models. As we claimed in category definition, world simulators are models that capable of generating video content that adhering to physical rules and aligning with the executed actions. The methods and experiments in our paper were specifically designed to achieve this goal. Expiclict Perceptual evaluation for physical rules adherence and Implicit manipulation evaluation for action executability.
>
> **Addressing Human Feedback Annotations**
> You mentioned, *"the article should focus more on how to better utilize human feedback annotations to improve the downstream video-to-action problem."* This suggestion stems from a misunderstanding of our paper’s scope. Since our work centers on evaluating the capabilities of video generation models, the primary focus is to leverage the HF-Embodied dataset to train scoring models and analyze the evaluation results for video models, rather than to improve downstream task performance. These aspects are analyzed thoroughly in our paper.
>
> **Acknowledgment and Future Work**
> We appreciate your recognition of the HF-Embodied dataset’s contributions. While its primary role in this work is for evaluation, we agree that it holds immense potential for enhancing video generation models and improving the ability of world simulators to execute embodied downstream tasks. We have incorporated a discussion of these possibilities into the revised version of the paper in Section B.1 in Appendix, highlighting its broader applications and future prospects.
>
> ---
>
> In conclusion, we hope this clarification resolves the misunderstanding and provides a clearer context for evaluating the paper. The focus of our work lies in establishing a novel evaluation framework for world simulators, addressing a critical gap in predictive model assessment. We are grateful for your thoughtful feedback, which has helped refine our presentation.

---

> > ### Comment · Reviewer_kNpR · 2024-11-28
> > **Response to Author**
> >
> > Thank you for your clarification.
> >
> > Regarding your mention of "evaluating and analyzing the capability of video generation models as World Simulators, addressing the limitations of traditional video evaluation methods in assessing attributes unique to world models," I do not completely agree.
> >
> > For example, in the embodiment task:
> >
> > There are many works on hierarchical evaluation, such as UniPi[1], Robodreamer[2], UniSim[3], etc.
> > Regarding explicit evaluation, papers like Robodreamer[2] and UniSim[3] also emphasize analyzing the generation quality. If the difference in this paper's evaluation compared to such works lies in assessing human preference, then this distinction depends on data annotation. This innovation is insufficient if human preference evaluation is not further empowered to enhance downstream tasks.
> >
> > In summary, I emphasize the connection with downstream tasks because there have been many similar works on dual evaluation schemes previously, and the only difference between the authors and prior works is the method of explicit evaluation. However, I do not see how this explicit evaluation method benefits downstream tasks. If it does not, merely adding human preference has limited significance.
> >
> > Additionally, existing video diffusion models do not adequately explain physical rules. If this point is to be addressed, the paper should reference [4] more extensively to provide additional analysis on physical rules, as this is crucial for the embodiment task.
> >
> >
> > [1] UniPi: Learning universal policies via text-guided video generation
> > [2] RoboDreamer: Learning Compositional World Models for Robot Imagination
> > [3] Learning Interactive Real-World Simulators
> > [4] How Far is Video Generation from World Model: A Physical Law Perspective

---

> ### Author Response · Authors · 2024-11-21
> **Response to kNpR (2/3)**
>
> ### Weakness 1: Discussion about UniPi.
>
> **A:**  We appreciate the reviewer’s detailed feedback and would like to clarify a potential misunderstanding about the scope and methodology of our work.
>
> **Clarification of Scope and Methodology**
> Our primary goal is to evaluate video generation models as world simulators, not to propose new downstream task paradigms or policies. To ensure that the action policies used in our simulation scenarios are effective, we chose the policy framework from SuSIE, which has been validated as a highly efficient and robust policy for the specific tasks we evaluate. UniPi framework poses challenges as it is less aligned with our evaluation framework, potentially compromising fairness and consistency in comparing generative video models as world simulators.
>
> **Supporting Evidence**
> As expected, the results confirm that SuSIE’s policies provide better alignment with our evaluation framework, as shown in the updated Tab.4 in Appendix (referencing performance results from SuSIE and UniPi). This further validates our methodological choice and highlights the necessity of selecting policies that ensure the evaluation remains consistent and meaningful.
>
> ---
>
> ### Weakness 2: More trajectories and comparison with SOTA methods.
>
> **A:**  We appreciate the reviewer’s valuable feedback and concerns about the evaluation metrics. Here, we address these points in detail while clarifying our methodological decisions and objectives.
>
> **Clarification of Evaluation Methodology**
> Our primary focus in this paper is evaluating video generation models as world simulators, not developing new downstream video-to-action (Video2Action) models or comparing against state-of-the-art (SOTA) approaches in specific tasks like CALVIN. To ensure fairness and consistency in our evaluations, we used a validated Video2Action model derived from SuSIE's framework. This choice aligns with our paper's goal of providing a level playing field to compare the generative capabilities of different video models without introducing variability from alternative or more complex Video2Action models.
>
> While SOTA methods or novel Video2Action models could potentially improve individual task performance, evaluating their efficacy is outside the scope of this work. Instead, our primary contribution is establishing an evaluation framework that highlights the capabilities of video generation models in embodied tasks.
>
> **Additional Experiments for Credibility**
> The reason we conducted 20 tests was to accelerate the evaluation process, a strategy also employed in SuSIE, where they used 25 tests to evaluate the video-based UniPi model. These details can be found in Section B of the Appendix in SuSIE [1].
>
> To address concerns about the limited number of evaluations, we conducted additional experiments using 100 tests in the following table. The results, also summarized in the updated Tab.4 in Appendix, show minimal deviation from the trends observed with 20 runs, further validating our findings. These results reinforce that our evaluation pipeline reliably assesses the capabilities of video generation models as world simulators. This discussion has also been incorporated into the revised paper.
>
> **On SOTA Comparisons**
> The reviewer raises a valid point about the absence of comparisons with SOTA methods. However, incorporating SOTA Video2Action models into our pipeline would deviate from our primary goal of providing a fair comparison of generative video models as world simulators. Exploring SOTA integration is indeed a meaningful direction for future work, but it lies outside the scope of our current objectives. We have incorporated the additional experimental results and detailed discussions in the revised manuscript, along with acknowledgment of the reviewer’s suggestions as avenues for future exploration.
>
> | Method              | Task completed in a row (%) ↑ | 1    | 2    | 3    | 4    | 5    | Avg. Len. ↑ |
> |---------------------|-------------------------------|------|------|------|------|------|-------------|
> | UniPi* (HiP)        |                               | 0.08 | 0.04 | 0.00 | 0.00 | 0.00 |     -           |
> | UniPi* (Susie)      |                               | 0.56 | 0.16 | 0.08 | 0.08 | 0.04 |     -           |
> | Open-Sora-Plan      |                               | 0.89 | 0.72 | 0.63 | 0.34 | 0.32 |     3.12        |
> | DynamiCrafter       |                               | 0.93 | 0.69 | 0.51 | 0.27 | 0.18 |     2.64        |
> | EasyAnimate         |                               | 0.92 | 0.55 | 0.32 | 0.16 | 0.13 |     2.08        |
>
> **Table Caption:** *Detail Result of Robot Manipulation in Implicit Manipulative Evaluation, by running 100 trajectories, * reported by Susie.
>
> ###Reference
> [1] Black K, Nakamoto M, Atreya P, et al. Zero-shot robotic manipulation with pretrained image-editing diffusion models[J]. arXiv preprint  arXiv:2310.10639, 2023.

---

> ### Author Response · Authors · 2024-11-21
> **Response to kNpR (3/3)**
>
> ### Weakness 3: Misunderstanding of valuable aspects.
> **A:** We appreciate the reviewer’s feedback and the recognition of the value of our human feedback annotations. However, we would like to clarify some misunderstandings regarding the motivation and contributions of our work.
>
> The primary goal of our paper is to evaluate the capability of video generation models as world simulators, addressing the challenges of assessing physical rules and logical consistency in video evaluation. To achieve this, we designed two novel evaluation methods that go beyond traditional score-based approaches, enabling us to effectively evaluate these critical attributes. The introduction of human feedback annotations plays a key role in this process. Through carefully designed evaluation metrics, curated video data, detailed manual annotations, and trained scoring models, we provide a comprehensive assessment of video models across multiple dimensions, such as adherence to physical rules and logical consistency.
>
> We also appreciate your insightful comments regarding the broader potential applications of the dataset. While in this work the dataset's primary role is to support the evaluation of video models, as you mentioned, the human feedback annotations could indeed be leveraged in future research. For instance, Reinforcement Learning from Human Feedback (RLHF) could use this dataset to further enhance video model capabilities, or it could help improve the performance of world simulators in downstream embodied intelligence tasks.
>
> **Revision and Future Work:**
> In light of your suggestion, we have revised the manuscript to include a more comprehensive summary of the dataset's contributions and its potential applications beyond evaluation. Specifically, we have discussed the possibilities for future work, such as using the dataset for RLHF and exploring its role in enhancing world simulators' performance in embodied tasks. We believe these additions will provide a clearer understanding of the dataset's broader utility while maintaining focus on our work's primary objective.
>
> ---
>
> ### Weakness 4: Misunderstandings have led to doubts regarding the generalization and capabilities of our framework.
> **A:** We appreciate the reviewer's feedback and would like to clarify some misunderstandings regarding our framework's motivation and contributions.
>
> First, Figure 3 represents the proposed evaluation pipeline, which is designed to provide a fair and consistent evaluation process for downstream tasks. Its purpose is not to improve downstream task performance but to ensure an objective assessment of video generation models as world simulators. The core contribution of our framework lies in analyzing the capabilities of video generation models as world simulators by identifying the aspects in which they meet or fall short of the requirements and highlighting the differences between various models. We do not claim that the improved evaluation results from Open-Sora are due to our framework. Therefore, the statement, "This indicates that the framework's design is not advancing the task further; it merely replaces it with a better video generation model, which still stems from the performance improvements of Open-Sora," reflects a misunderstanding of our framework's motivation and contributions.
>
> Second, regarding Figure 3 and the task structure, the CALVIN dataset categorizes tasks into five levels based on the number of instructions (e.g., Level 4 tasks require four instruction steps). Similarly, for Minecraft and autonomous driving scenarios, we adopted an offline task decomposition approach, where the execution steps for each task were predefined and fixed before downstream evaluation. This approach was chosen to ensure fairness in evaluating video generation models, as all models are tested with the same task prompts, avoiding potential biases caused by online task decomposition. Additionally, this design accelerates the downstream evaluation process. It is essential to emphasize that our pipeline was designed to evaluate video generation models, not to enhance their downstream task performance. Hence, the critique regarding "solving practical problems, traditional methods are still employed" does not align with the goals or scope of our work.
>
> Our framework is a fair and reasonable evaluation strategy aimed at providing meaningful insights into the strengths and limitations of video generation models as world simulators, rather than a pipeline designed for solving practical tasks or improving downstream capabilities.

---

> ### Author Response · Authors · 2024-11-25
> **Official Comment by Authors**
>
> We hope this message finds you well. We have noted the deadline for open discussion of ICLR 2025 is approaching, yet we have not yet received any feedback from you. In light of this, we sincerely wish to know if we can receive any updated comments regarding to our submission 873, titled "WorldSimBench: Towards Video Generation Models as World Simulators". We are very pleased to hear from you on the reviewer’s comments.

---

> ### Comment · Reviewer_kNpR · 2024-11-28
> **Response to Author**
>
> As for the other issues, I thank the authors for addressing some concerns regarding the CALVIN experiment's test sequences.
>
> However, other problems remain.
>
> 1. Your robot dataset is RH20T-P (real-world) , but the experiments were conducted in the simulation environment of CALVIN. You should provide results from the real scenarios.
> 2. Figure 3 appears to be quite general, but you adopted an offline task decomposition approach in the actual experiments. I believe the consistency between the method and the experiments is lacking because, in practice, one must use a Task Planner for decomposition, which aligns with real-world scenarios. Simplifying this operation is unreasonable.
> 3. If the primary purpose of this work is to evaluate video generation models, then the innovation is quite limited. There are many methods for evaluating video generation models; the integration with downstream tasks adds significance to this paper. One of your innovations is HF-Embodiment, and the combination with the Embodiment task distinguishes your work from many others in video generation. Therefore, I hope the authors elaborate on the Embodiment section, mainly on how the human preference annotation data strengthens the Embodiment task as a focal point. This distinguishes your work from others.

---

> > ### Author Response · Authors · 2024-11-28
> > **Response to kNpR**
> >
> > Thank you for your constructive and thoughtful comments. They were indeed helpful in improving the paper. We take this opportunity to address your concerns:
> >
> > ### Q1:
> > I'm sorry for the misunderstanding regarding our evaluation method. The **HF-Embodied Dataset** is used exclusively in the **Explicit Perceptual Evaluation** component of our evaluation framework, and is not utilized in the **Implicit Manipulative Evaluation**. **In the Implicit Manipulative Evaluation, we chose a reproducible simulation platform, which has been used by multiple works [1][2][3], for closed-loop evaluation.** Additionally, based on the capabilities of current video generation models, we performed additional fine-tuning on the CALVIN task to ensure the quality of the generated videos, thereby obtaining meaningful downstream task evaluation results.
> >
> > ### Q2:
> > I'm sorry again for the misunderstanding regarding our work's focus. As stated in the caption of **Figure 3**: "_Overview of Implicit Manipulative Evaluation_," **it demonstrates the evaluation framework, not the method for executing embodied tasks.** As mentioned in the response to **kNpR (3/3)** titled "Misunderstandings have led to doubts regarding the generalization and capabilities of our framework," we conduct offline decomposition to ensure that the factors affecting downstream task success rates are solely attributed to the video generation model. **If we were to perform online decomposition, task decomposition inconsistency due to the instability of LLMs would impact task success rates, severely affecting the fairness of closed-loop evaluation for video generation models.** Therefore, offline decomposition is necessary for video evaluation tasks.
> >
> > ### Q3:
> > Here, we would like to reiterate our innovation: **we are the first work to evaluate **S3-level** predictive models. By employing **Implicit Manipulative Evaluation** and **Explicit Perceptual Evaluation**, we overcome the shortcomings of S2-level evaluation methods, which are unable to effectively assess the attributes of S3-level models.** Based on the evaluation results, we also analyze the current possibilities and limitations of video generation models as world simulators. As you mentioned, "One of your innovations is HF-Embodiment," we have provided both qualitative and quantitative experiments that demonstrate the validity and significance of our HF-Embodied Dataset for evaluating video generation models. **We would like to kindly remind you that the HF-Embodied Dataset is not only meaningful for improving embodied task success rates, but it is also a critical tool for effectively evaluating video generation models and addressing the challenges of S3-level model evaluation, which traditional score-based methods cannot solve.** This is of significant importance both in the field of video generation and the world model domain.
> >
> > Thank you for your valuable time and effort in reviewing our work. We would greatly appreciate receiving your feedback on our response to facilitate further discussion. If any aspects of our explanation are unclear, please feel free to let us know. Thank you once again for your invaluable comments and consideration, which are greatly beneficial in improving our paper.
> >
> > [1]Zero-Shot Robotic Manipulation with Pretrained Image-Editing Diffusion Models (ICLR2024)
> >
> > [2]Closed-Loop Visuomotor Control with Generative Expectation for Robotic Manipulation (NeurIPS 2024)
> >
> > [3]VidMan: Exploiting Implicit Dynamics from Video Diffusion Model for Effective Robot Manipulation (NeurIPS 2024)

---

> ### Author Response · Authors · 2024-11-28
> **Response to kNpR**
>
> Thank you for your constructive and thoughtful comments. They were indeed helpful in improving the paper. We take this opportunity to address your concerns:
>
> ### Innovation:
>
> Here, **I would like to reaffirm that our focus is on the evaluation of video generation models**. We employ specially designed Explicit Perceptual Evaluation and Implicit Manipulative Evaluation to assess the logical capabilities and physical rule generation abilities of multiple existing video models across various embodied intelligence domains. We then analyze and discuss the results.
>
> On the other hand, UniPi [1], Robodreamer [2], and UniSim [3] aim to design reasonable world simulators to solve downstream embodied tasks, which is unrelated to the evaluation of video generation models. Their discussions focus on how to design frameworks to solve downstream tasks via video generation. I believe it is not appropriate to compare benchmarks and methodologies in this context.
>
> Additionally, in their papers, **the simple visual-level evaluation of diffusion models relies only on score-based global metrics like **FVD**, which fail to capture the local information within the videos.** Local information is crucial as it often reflects key attributes such as physical rules (e.g., the interaction between a robotic arm and an object is a key region, while large portions of the background are not critical). Therefore, we constructed the HF-Embodied Dataset and trained the corresponding scoring model. The scoring model provides targeted evaluations based on carefully designed dimensions (e.g., interaction, trajectory), effectively addressing tasks that traditional score-based models cannot complete. In our paper, we have provided both qualitative and quantitative analyses to demonstrate that the dataset and evaluation model can effectively assess video generation models, thus achieving the paper's motivation.
>
> Regarding your comment: _“However, I do not see how this explicit evaluation method benefits downstream tasks,”_ we believe this is a valuable research direction, but for the purpose of evaluating video generation, **it clearly falls outside the scope of our discussion.** We also hope that future researchers will utilize our dataset and models to construct state-of-the-art robotic frameworks for downstream tasks.
>
> ### Impossible Comparison:
>
> The work you referred to [4] is indeed meaningful, but you may have overlooked the timing of its publication on arxiv. Our paper was submitted to ICLR before the ICLR submission deadline on **October 1, 2024**, while [4] was published on **arXiv on November 4, 2024**, just 20 days ago. Therefore, **it is not possible to reference a paper that was made publicly available after our submission.** Furthermore, since [4] has not yet been peer-reviewed, it may not be suitable for further comparison at this time. Nonetheless, we sincerely appreciate your contribution to improving our paper.
>
> [1] UniPi: Learning universal policies via text-guided video generation
>
> [2] RoboDreamer: Learning Compositional World Models for Robot Imagination
>
> [3] Learning Interactive Real-World Simulators
>
> [4] How Far is Video Generation from World Model: A Physical Law Perspective

---

> > ### Comment · Reviewer_kNpR · 2024-11-30
> > **Response to Author**
> >
> > Thank you very much for your response.
> >
> > 1. As mentioned in the quoted article, this paper is not the first to use the dual evaluation framework. The difference between this paper and previous works lies in introducing a better evaluation method.
> >
> > 2. I acknowledge that this evaluation method is more beneficial than earlier approaches; however, since this evaluation method has not been applied to downstream tasks, the overall innovativeness is limited.
> >
> > 3. Even regarding the evaluation scheme you proposed, it still does not capture physical rules. For embodied tasks, physical rules should focus more on rules involving fundamental physical principles. Notably, my mention of [1] is not to suggest that you should compare from the experiment but rather to indicate that the paper should consider more elements related to embodiment (such as force, touch, collision, etc.) rather than just remaining at the so-called physical rules in terms of video generation.
> >
> > In summary, I appreciate the substantial experiments and efforts made by the authors, which is why I have given a score of 5. However, I believe the current work is incomplete and does not truly advance embodied tasks; thus, its contribution remains limited. I will maintain my score.
> >
> > [1] How Far is Video Generation from World Model: A Physical Law Perspective

---

> > > ### Author Response · Authors · 2024-12-01
> > > **Response to kNpR**
> > >
> > > ### Q1: Clarification on Our Evaluation Framework
> > > Our primary goal has always been to evaluate video generation models from the perspective of embodied intelligence. We aim to address the shortcomings of previous video evaluation methods by designing a reasonable video evaluation framework that aligns with the properties of world models. The frameworks you mentioned, which focus on solving downstream embodied tasks, are not designed for video model evaluation. Their objective is to demonstrate the effectiveness and interpretability of their frameworks in tackling downstream practical tasks.
> > >
> > > As you noted, these methods utilize online task decomposition since their goal is to ensure task completion rather than fairly comparing video generation models. To date, no prior work has evaluated multiple video generation models and analyzed their results. We hope you approach our work from an evaluation standpoint rather than as a method for solving embodied tasks. Please consider the motivation, superiority, and valuable results and analyses our work offers in the context of video model evaluation.
> > >
> > > ### Q2: Human Feedback Dataset for Downstream Tasks
> > > We appreciate your valuable suggestions from the perspective of solving downstream embodied tasks, though it is beyond the scope of our current evaluation work. Nevertheless, we agree with your viewpoint. Exploring how human feedback datasets could promote embodied evaluation models for specific tasks is a meaningful direction.
> > >
> > > Based on our experimental structure, we propose possible suggestions for such research:
> > > 1. Develop an effective RLHF model based on video generation.
> > > 2. Design a corresponding reward model tailored to the RLHF framework (note that our evaluation model is designed for assessment and would require significant redesign and validation to serve as a reward model).
> > > 3. Establish specific benchmarks to verify how human feedback improves downstream tasks and analyze the nature of these improvements in different scenarios.
> > >
> > > This research is both necessary and promising. However, each step requires substantial computational and human resources, necessitating an independent study.
> > >
> > > ### Q3: Discussion on [1]
> > > Thank you for your suggestion regarding [1]. We have carefully reviewed and analyzed the paper:
> > > > "[1] provides a systematic investigation of video diffusion models from a physical law perspective, supported by extensive experiments and solid analysis in 2D space, showing several intriguing findings and improving the understanding of physical laws in video diffusion models."
> > >
> > > Here is a summary addressing your points:
> > > 1. **Evaluation Perspective**: [1] simplifies physical rules to a 2D plane to test various properties. In contrast, we evaluate from the perspective of embodied intelligence, emphasizing the complexity of real-world scenarios. To ensure that the results of our Explicit Perceptual Evaluation reflect real-world conditions, we deliberately avoided simplifying the environment.
> > > 2. **Evaluation Dimensions**: Your comment, "the paper should consider more elements related to embodiment (such as force, touch, collision, etc.) rather than just remaining at the so-called physical rules in terms of video generation," suggests a misunderstanding. If you review the Evaluation Dimension section of our Explicit Perceptual Evaluation (Table 3 in the main text), you will find that we evaluate physical attributes such as touch (Embodied Interaction), trajectory, and velocity. Additionally, we include attributes unique to real-world scenarios, like perspectivity, which are absent in 2D settings. These attributes are critical for evaluating the capability of video models as world simulators and represent a significant contribution of our work.
> > >
> > >
> > > [1] How Far is Video Generation from World Model: A Physical Law Perspective
> > >
> > > ---
> > >
> > > If our answers and explanations about the contribution of our study to evaluating video generation models as world simulators address your concerns, we sincerely hope that you will consider raising your **rating**. If you still have any doubts, we would be more than happy to discuss them further with you. We look forward to collaboratively advancing the fields of world modeling and video generation.

---

### Official Review · Reviewer_jqzP · 2024-11-04

**Soundness:** 4
**Presentation:** 3
**Contribution:** 3
**Rating:** 8
**Confidence:** 4

**Summary:**

This paper introduces a hierarchy for classifying predictive models based on their capabilities and level of embodiment. This categorization aims to advance research and development in the field and provides a framework for evaluating world simulators. A new evaluation framework called WorldSimBench is proposed, which includes two types of evaluations: explicit perceptual evaluation and implicit manipulative evaluation. The paper conducts extensive testing across multiple world simulator models and analyzes the results in detail. Further, A large dataset called HF-Embodied Dataset is developed, which includes human feedback on 3D environments across various scenarios and dimensions and enables the evaluation of world simulators and has broader applications in video generation models, such as alignment.

**Strengths:**

Explicit Perceptual Evaluation: This assesses the visual quality, condition consistency, and embodiment of the generated content through human preference evaluation. The HF-Embodied Dataset, containing fine-grained human feedback, is used to train a Human Preference Evaluator.

Implicit Manipulative Evaluation: This assesses the World Simulator’s performance by converting the generated videos into control signals and evaluating their effectiveness in embodied tasks within three scenarios: Open-Ended Embodied Environment (OE), Autonomous Driving (AD), and Robot Manipulation (RM).

Comprehensive Evaluation: WorldSimBench provides a holistic assessment of World Simulators by considering both visual and action aspects, addressing the limitations of existing benchmarks that focus only on aesthetic quality or task completion.

Human-Centric Evaluation: The use of human preference evaluation and fine-grained feedback allows for a more intuitive and accurate reflection of the quality and characteristics of the generated videos, including their adherence to physical rules.

Diverse Scenarios: The evaluation covers three representative embodied scenarios, providing a comprehensive benchmark for evaluating World Simulators across a range of real-world tasks.

Open-Source Dataset: The HF-Embodied Dataset is open-source and can be used for various applications beyond evaluating World Simulators, such as alignment and other video generation tasks.

**Weaknesses:**

Limited Scope: The evaluation primarily focuses on physical rules and 3D content within the context of embodied intelligence, specifically robots. The evaluation of World Simulators in other scenarios with more complex physical rules and diverse entities requires further exploration.

Static Evaluation Limitations: The evaluation of World Simulators using static benchmarks may not fully capture their dynamic nature and real-time requirements in interactive environments. Further research is needed to develop more effective evaluation methods for dynamic scenarios.

Limited Generalizability of Video-to-Action Models: The performance of video-to-action models in converting generated videos into control signals may vary depending on the specific task and environment. Developing more robust and generalizable video-to-action models is crucial for the practical application of World Simulators.

**Questions:**

Could the generated videos help improve the performance of VLA models?

---

> ### Author Response · Authors · 2024-11-21
> **Response to jqzP (1/2)**
>
> Thank you for your constructive and thoughtful comments. They were indeed helpful in improving the paper. We take this opportunity to address your concerns:
>
> ### List of changes in the manuscript:
> 1. Section G in Appendix. Discussions about Video Generation Model and VLA.
>
> ---
>
> ### Weakness 1: Limited Scope.
>
> **A:**  Thank you for pointing out the scope of our evaluation framework. We would like to emphasize that our work is the first to propose a comprehensive "how-to" evaluation methodology for video generation models as world simulators, providing a robust and reasonable evaluation pipeline.
>
> While our current evaluation focuses on scenarios involving physical rules and 3D content within embodied intelligence, the proposed framework is inherently extensible. For instance, it can be adapted to evaluate world simulators in broader contexts, such as modeling meteorological phenomena (e.g., cloud formations and weather changes) or representing chemical reactions. These domains, while requiring specialized domain knowledge and curated datasets, can directly leverage our pipeline for assessment and further exploration.
>
> By following our evaluation approach, researchers can tailor the pipeline to meet the requirements of specific applications across various domains. This demonstrates the flexibility and adaptability of our framework, enabling its use for a wide range of world simulator evaluations beyond the current scope. We hope this addresses your concern and highlights the potential impact of our work.
>
> ---
>
> ### Weakness 2: Static Evaluation Limitation.
>
> **A:**  Thank you for highlighting this important aspect. We would like to clarify that our evaluation framework is far from being static. On the contrary, one of the unique strengths of our work is its focus on dynamic evaluation, which sets it apart from traditional static benchmark approaches.
>
> Our downstream tasks are designed to capture the dynamic nature and real-time requirements of interactive environments. For instance:
> 1. **Open-Ended Embodied Environment (OE):** In this setting, agents interact with their surroundings in a continuous, open-ended manner, requiring the generated videos to adhere to physical laws and maintain logical coherence over time.
> 2. **Autonomous Driving (AD):** This scenario involves predicting future frames for a vehicle navigating dynamic environments, where real-time decision-making and trajectory consistency are critical.
> 3. **Robotic Manipulation (RM):** Here, agents must predict and execute a sequence of actions to manipulate objects, with tasks such as object stacking or rearranging relying on accurate, time-sensitive video generation.
>
> These scenarios emphasize the temporal consistency and interaction dynamics between the agent and the environment, offering a more comprehensive and realistic evaluation compared to static benchmarks. Our framework not only evaluates visual quality but also assesses how well the generated content supports successful task execution in these dynamic and interactive settings.
>
> We believe this approach directly addresses the concerns regarding static evaluation and underscores the robustness of our pipeline in capturing the dynamic and real-time nature of world simulators.
>
> ---

---

> ### Author Response · Authors · 2024-11-21
> **Response to jqzP (2/2)**
>
> ### Weakness 3: Limited Generalizability of Video-to-Action Models.
>
> **A:**  Thank you for pointing this out. We would like to clarify that while Video-to-Action (Video2Action) models are indeed an important research area, they are not the focus of our study. Our work primarily investigates the evaluation of world simulators, and the Video2Action model serves as a tool within our pipeline rather than being tightly coupled with the simulator itself.
>
> We believe the use of Video2Action models in our evaluation framework is both reasonable and relevant for several key reasons:
> 1. **Groundtruth-Based Training:** All the Video2Action models used in our experiments are trained on video datasets with explicit action ground truth. This ensures that the models are appropriately equipped to generate control signals directly aligned with the underlying actions in the dataset.
> 2. **Evaluation Relevance:** Our Video2Action models are specifically designed to facilitate the assessment of world simulators. By converting generated videos into control signals, these models allow us to measure how well the simulated content supports downstream tasks, which is central to our evaluation framework.
> 3. **Alignment with Upstream Metrics:** The results derived from the Video2Action models show strong correlations with the perceptual evaluations of the upstream world simulator. This consistency validates the utility and relevance of Video2Action models in our pipeline and highlights their ability to reflect meaningful insights about the simulator’s performance.
>
> In summary, while Video2Action is a broader research domain, our use of it in this paper is well-justified and carefully scoped to support the evaluation of world simulators. It serves as a complementary component that enables us to bridge the gap between generated content and task-level performance, reinforcing the validity of our proposed evaluation methodology.
>
> ---
>
> ### Question: Discussion about whether Generated Videos Help Improve the Performance of Vision-Language-Action (VLA) Models.
>
> **A:**  The generated videos have the potential to significantly enhance the performance of Vision-Language-Action (VLA) models by addressing two key challenges in training such models: the availability of diverse, high-quality training data and the need for effective reward functions in real-world scenarios.
>
> 1. **Data Augmentation and Hindsight Relabeling for Imitation Learning**
> Generated videos can serve as a valuable source of synthetic data for training VLA models. By leveraging the diversity and scalability of generative models, we can create a wide array of training scenarios, covering edge cases and rare events that are difficult to capture in real-world datasets. Additionally, these videos enable hindsight relabeling, a process where we retrospectively adjust the labels of generated data to align with desired outcomes. This approach is particularly effective for imitation learning, allowing VLA models to learn optimal behavior by mimicking successful trajectories represented in the generated videos. By expanding the data distribution and improving its quality, generative videos can lead to more robust and generalizable VLA models.
>
> 2. **Reward Generation for Online Reinforcement Learning**
> Beyond data augmentation, generated videos can act as a Reward Generator in reinforcement learning (RL) contexts. Unlike traditional RL setups that rely on pre-defined reward functions within a simulator, generative videos enable the creation of dense and context-aware reward signals tailored to real-world tasks. For example, they can simulate desirable outcomes or intermediate goals, providing detailed feedback to the agent. This capability is particularly crucial for transferring RL models to real-world environments, where designing explicit reward functions is often impractical. By aligning the generated rewards with real-world objectives, we can bridge the gap between simulation and reality, allowing VLA models to achieve higher performance in real-world tasks.
>
> We have incorporated these discussions into the revised version of our paper in Section G in Appendix.

---

> ### Author Response · Authors · 2024-11-25
> **Official Comment by Authors**
>
> We hope this message finds you well. We have noted the deadline for open discussion of ICLR 2025 is approaching, yet we have not yet received any feedback from you. In light of this, we sincerely wish to know if we can receive any updated comments regarding to our submission 873, titled "WorldSimBench: Towards Video Generation Models as World Simulators". We are very pleased to hear from you on the reviewer’s comments.

---

> ### Author Response · Authors · 2024-12-01
> **Official Comment by Authors**
>
> Thank you for taking the time to review our paper and for your support for its acceptance. We hope our rebuttal enhances your confidence in our paper. If so, we wonder if this can be reflected in an increased confidence score. We would be happy to provide any additional materials if needed.

---

### Official Review · Reviewer_HLX1 · 2024-11-04

**Soundness:** 3
**Presentation:** 3
**Contribution:** 3
**Rating:** 5
**Confidence:** 4

**Summary:**

The training of video generation models in WorldSimBench involves fine-tuning multi-modal large models on the HF-Embodied Dataset, which provides multi-dimensional human feedback on visual quality, instruction alignment, and embodiment. Using techniques like LoRA for efficient parameter tuning, the models optimize for perceptual, instruction consistency, and embodiment losses to enhance their alignment with human preferences and physical consistency across various tasks. Human Preference Evaluator, trained on HF-Embodied Dataset, plays a key role by scoring generated videos and providing iterative feedback, enabling the models to improve continuously in producing realistic, instruction-compliant, and task-appropriate video outputs across scenarios like open-ended environments, autonomous driving, and robotic manipulation.

**Strengths:**

1. Comprehensive dual evaluation framework assessing models on both visual quality and action-level performance.

2. Detailed human feedback dataset (HF-Embodied Dataset) provides multi-dimensional evaluation criteria for realism and task alignment.

3. Efficient training approach with LoRA and Human Preference Evaluator enables targeted, adaptive improvements without extensive computational resources.

4. Versatile benchmarking across diverse scenarios (open-ended environments, autonomous driving, and robotic manipulation) to ensure robustness in varied real-world applications.

**Weaknesses:**

1. The data for the different scenarios was not trained together, potentially limiting the model's ability to generalize across multiple embodied contexts and requiring individual training processes for each environment, which could hinder efficiency and scalability.

2. There is a relative lack of qualitative results in the study, which would have provided richer insights into how well the models handle nuanced, complex interactions or specific visual challenges within each scenario, potentially limiting the interpretability of performance beyond numerical metrics.

3. Some contributions, such as integrating evaluations across different embodied scenarios, might feel more like a survey. While valuable, these elements primarily build on existing consensus in the field and might not introduce new methods or substantial innovations in evaluation techniques.

4. The role of the Human Preference Evaluator, though useful in aligning generated content with human preferences, has potential that remains underexplored. The paper mainly employs it for evaluation without exploring its broader applications. Additionally, it is not a general-purpose evaluator, as adapting it to a new dataset requires re-collecting data and retraining, which limits its flexibility and scalability.

**Questions:**

See Weaknesses

---

> ### Author Response · Authors · 2024-11-21
> **Response to HLX1 (1/2)**
>
> Thank you for your constructive and thoughtful comments. They were indeed helpful in improving the paper. We take this opportunity to address your concerns:
>
> ### List of changes in the manuscript:
> 1. Section B.1 in Appendix. Discussions about the future work of HF-Embodied dataset.
> 2. Section C.2 in Appendix. Qualitative Results and Analyses.
>
> ---
>
> ### Weakness 1: The model's separate training for different scenarios may limit generalization and hinder efficiency and scalability.
>
> **A:**  Thank you for raising this important point regarding the training of models across multiple scenarios. The current dominant setting in robotics and embodied AI tasks often involves a single model tailored for specific ontologies (or embodied agent types). Training across diverse scenarios simultaneously is indeed an emerging trend but remains technically challenging. As our work focuses on evaluation strategies, we followed the common practice of training models within their respective domains, which is consistent with current standards in the field.
>
> From a technical perspective, we did explore training across multiple embodied scenarios. However, the results were suboptimal, as the models failed to generate videos that met the minimum evaluation standards. We believe this reflects the limitations of current open-source generative models, which are still in the early stages of development and lack the capacity to generalize high-quality video generation across diverse domains.
>
> To ensure fairness and reliability in evaluation, we fine-tuned models separately for each embodied scenario before conducting assessments. This allowed us to fairly evaluate their performance within specific contexts. Your comment has also prompted us to reflect on broader challenges: even advanced models like GPT occasionally struggle with vertical domains, and fine-tuning with domain-specific data has proven effective in addressing these issues. Similarly, for generative world models, it may be necessary to establish a general-purpose pre-trained model combined with fine-tuning on specific domain data to ensure high performance across different embodied scenarios. This is a promising direction for future work.
>
> ---
>
> ### Weakness 2: Lack of qualitative results.
>
> **A:**  Thank you for pointing this out. We agree that qualitative results can provide deeper insights into the models’ performance, particularly in how they handle nuanced, complex interactions or specific visual challenges within each scenario. In our revised manuscript, we include a qualitative analysis of generated videos under the three embodied scenarios. Each video is represented by three evenly sampled frames, with the corresponding generation instructions listed above the video. To the left of the videos, we provide the scores of the key embodied attributes labeled by human preference evaluator. More visualization results and qualitative analyses can be found in the revised paper, Section C.2 in the Appendix.
>
> ---
>
> ### Weakness 3: Innovation of integration of evaluations across scenarios.
>
> **A:**  Thank you for your feedback regarding the perceived lack of substantial innovation in our evaluation framework. We would like to emphasize the novelty and significance of our contributions, as they address a critical challenge in the field: how to evaluate video generation models as world simulators.
>
> 1. **Novel Evaluation Framework:**
>    To the best of our knowledge, this is the first framework specifically designed to evaluate video generation models as world simulators. Our framework comprises two complementary components:
>    - **Human-aligned perceptual evaluation**, which ensures alignment with human cognitive judgments.
>    - **Physics-driven embodied task evaluation**, which steps beyond human perception to assess the rationality of the generated content in real physical environments. This dual approach not only addresses the limitations of traditional score-based evaluations but also uncovers potential consistencies between perceptual alignment and task-level performance, validating the effectiveness and reliability of our evaluation strategy.
>
> 2. **Taxonomy and Embodied Tasks:**
>    In the perceptual alignment evaluation, we introduced a well-defined taxonomy to rigorously categorize human perceptual dimensions. Additionally, we evaluated downstream embodied tasks across diverse scenarios, covering multiple embodied settings, to ensure a comprehensive assessment of the models’ capabilities.
>
> Both components are designed to solve the central question of how to evaluate video generation models as world simulators. We believe these contributions are significant and provide valuable tools for advancing research in this area.
>
> ---

---

> ### Author Response · Authors · 2024-11-21
> **Response to HLX1 (2/2)**
>
> ### Weakness 4.1: Discussion the future work of Human Preference Evaluator.
>
> **A:**  Thank you for your insightful suggestion regarding the broader potential of the Human Preference Evaluator. While we agree that it could be used for tasks like RLHF (Reinforcement Learning with Human Feedback) to enhance model performance, our primary focus in this work is on evaluation rather than optimization. Exploring such applications would extend beyond the scope of this paper.
>
> That said, we acknowledge the importance of discussing the broader applicability of this evaluator. In the revised version of our paper, we have added a more comprehensive discussion in the Section B.1 in Appendix about how this dataset and evaluation framework could inspire advancements in aligning models with human preferences, including its potential use in improving generation quality. We hope this addresses your concern and highlights the flexibility and value of the proposed approach.
>
> ---
>
> ### Weakness 4.2: Lack of flexibility and scalability due to dataset-specific retraining.
>
> **A:**  Thank you for highlighting this important point regarding the limitations of our evaluator's generalizability. We would like to clarify that the primary contribution of our work is not to improve the generalization ability of video generation models but to propose a framework for evaluating such models effectively.
>
> Our focus is on validating the feasibility of our evaluation methodology, which we have demonstrated through experiments across three datasets with significant domain gaps. These experiments underscore the necessity of training on multiple datasets for fair and robust evaluation. While we recognize and agree with your perspective that a general-purpose evaluation model would be valuable, achieving this requires addressing several significant challenges:
>
> 1. **Powerful Base Models:** A stronger foundational model is essential to support general-purpose evaluation capabilities.
> 2. **Expanded Data Scale:** Increasing the scale and diversity of training data would be critical to improving generalization.
> 3. **Resource Intensiveness:** Developing such a model would require substantial computational and human resources, with each step demanding independent research efforts.
>
> We believe this is a meaningful and necessary direction for future work, and we have included suggestions along these lines in the revised version of our paper. However, we also want to emphasize that building a general-purpose evaluator lies beyond the scope of this work, as our contribution is centered on defining and validating evaluation strategies for video generation models as world simulators.

---

> ### Author Response · Authors · 2024-11-25
> **Official Comment by Authors**
>
> We hope this message finds you well. We have noted the deadline for open discussion of ICLR 2025 is approaching, yet we have not yet received any feedback from you. In light of this, we sincerely wish to know if we can receive any updated comments regarding to our submission 873, titled "WorldSimBench: Towards Video Generation Models as World Simulators". We are very pleased to hear from you on the reviewer’s comments.

---

> > ### Comment · Reviewer_HLX1 · 2024-11-26
> >
> > Thank you for your detailed response. I appreciate the clarifications and revisions you have made.
> >
> > 1. **Category Discussion**: I believe the discussion on the taxonomy section lacks depth and appears somewhat abrupt. For example, the paper mentions that
> >
> >     "*The rapidly evolving field of World Simulators offers exciting opportunities for advancing Artificial General Intelligence, with significant potential to enhance human productivity and creativity, especially in embodied intelligence. Therefore, conducting a comprehensive embodied evaluation of World Simulators is crucial,*"
> >
> >     It does not connect the taxonomy to its relevance for evaluation. Specifically:
> >    - How do the proposed categories enhance the evaluation process?
> >    - Are there specific benefits to introducing these categories for model training or evaluation that could be demonstrated with examples or evidence?
> >
> > 2. **Justification for Dual Evaluation Framework and HF-Embodied Dataset**: I think further justification is needed. The broader community will likely validate the Implicit Manipulative Evaluation, so my main concern lies with the **Explicit Perceptual Evaluation**. You mentioned training a score model using the HF-Embodied Dataset, but the quality of the dataset has not been validated, making it difficult to further assess the effectiveness of the score model. For example:
> >    - Is there evidence that this dataset improves the performance of trained models?
> >    - Can it be demonstrated that outputs rated highly by the score model indeed correlate with better video quality or stronger alignment with the input text?
> >
> >     I fully understand that the focus of this paper is on how to evaluate world models and that it introduces a novel dataset and score model. However, I have not seen effective validation for these two components, as mentioned in reviewer kNpR's question.
> >
> > I appreciate that you have addressed my other concerns.

---

> ### Author Response · Authors · 2024-11-27
> **Response to: Category Discussion**
>
> We sincerely appreciate the reviewer's insightful comments and suggestions. We have taken this opportunity to address the concerns raised regarding the taxonomy discussion in the paper.
>
> ## Clarification of Motivation and Contribution
>
> We agree that the discussion on the taxonomy could benefit from more depth and clearer connections to the evaluation process. In the revised manuscript, we have elaborated on the motivation behind our taxonomy and how it directly enhances the evaluation of world simulators. Here's a more detailed explanation:
>
> ### S0-S2 versus S3: A Gradual Shift Towards Real-World Interaction
>
> The **S0-S2** categories primarily focus on models that operate within the digital world, where interactions are governed by simpler rules. These models predict and simulate within a controlled environment:
> - **S0**: Basic predictions based on non-visual abstract representations (e.g., predicting the future content with text description).
> - **S1**: More complexity, involving visual representations of goal frames but lacking process information.
> - **S2**: Models that visually describe the process from the present state to the future state, but ignore the logical and physical rules of how things develop in the real world.
>
> However, when we move to **S3 models**, which aim to interact with the real world (or realistic simulations of it), we encounter a need for a more sophisticated approach. **S3 models** must handle more complex dynamics, including real-time decision-making, reasoning about physical constraints, and adapting to dynamic environments. For example:
> - In **S2**, a video generation model may generate realistic frames (e.g., generating an image of a ball moving across a table) within a controlled environment.
> - However, an **S3 model** in dynamic environments like autonomous driving or robotic manipulation must deal with varying terrains, traffic, and the unpredictable behavior of pedestrians, all while adhering to physical constraints such as laws of motion and collision avoidance.
>
> ## The Need for a Hierarchical Evaluation Framework
>
> In terms of evaluation, there has not been a clear and widely accepted framework for assessing models in the **S3** category. By introducing the **S0-S3 taxonomy**, we provide a structured way to assess the different levels of predictive models, from simpler predictive models to complex, real-world interactions. This hierarchical structure allows us to develop specific evaluation criteria to assess world simulators.
>
> By differentiating the focus of **S0-S2** from that of **S3**, we can design a targeted evaluation framework and taxonomy that accurately assess the unique capabilities and relevant properties of world models in the **S3** category. For example:
> - **S2 models** may be evaluated on how well they generate the video.
> - **S3 models** need to be evaluated on their ability to handle more dynamic and unpredictable interactions, such as motion, traffic flow, or interaction results (e.g., object deformation or collision).
>
> ## Evaluating Models in the S3 Category
>
> To address the challenges of evaluating models in the **S3** category, we propose two innovative evaluation methods that go beyond the functionality of **S2** models. These methods aim to evaluate the unique capabilities of **S3** models:
> 1. **Explicit Perceptual Evaluation**: This method directly evaluates whether the generated content adheres to physical rules, logical consistency, and real-world constraints from human perception. For example, in a robotic manipulation task, the model’s generated video would be assessed on how well it simulates the physical laws governing object manipulation, such as trajectory and force.
> 2. **Implicit Manipulative Evaluation**: This method maps specific attributes (e.g., trajectory quality, speed) to real-world tasks that require those attributes. For instance, in the robotic arm scenario, the ability of the arm to manipulate an object in a physically accurate manner (e.g., not deforming rigid objects) can be evaluated in this way.
>
> These evaluation methods were designed to assess whether the model’s output not only looks realistic but also behaves realistically according to real-world rules, as **S3 models** are expected to possess more advanced capabilities.
>
> ## Conclusion
>
> We believe that the revised explanation of the taxonomy and its connection to the evaluation process addresses the reviewer’s concern about the depth of the discussion. The **S0-S3** hierarchy provides a valuable framework for understanding and evaluating the capabilities of world simulators, with specific benefits for model training and evaluation. By using this framework, we can create more targeted evaluation criteria suited to the distinct characteristics of models at each level, ensuring a fairer and more meaningful evaluation of their capabilities. We will revise the manuscript to include these clarifications.

---

> ### Author Response · Authors · 2024-11-27
> **Response to: Explicit Perceptual Evaluation**
>
> We appreciate your understanding that our focus is on evaluation. Here, we want to clarify the justification for the **Explicit Perceptual Evaluation**.
>
> We would like to emphasize that the **HF-Embodied Dataset** is a crucial component of our evaluation methodology. It is human-annotated based on clearly defined dimensions, allowing for an assessment of video quality from a perceptual standpoint. While it is true that human annotations may introduce some degree of subjectivity, we mitigate this concern by ensuring cross-validation among multiple annotators for each video, which helps maintain the consistency and reliability of the scores. Additionally, the annotations are grounded in explicit, well-defined criteria for each dimension, ensuring that the scoring process is rigorous and reproducible.
>
> ## Quantitative Results
>
> We understand the reviewer’s concern regarding the lack of validation for the quality of the **Human Preference Evaluator (HPE)**. To address this, we point to the results in **Table 3** and **Table 6**, where we demonstrate the alignment of the HPE and GPT model with human preferences. Both models are tasked with scoring a video based on specific dimensions (e.g., physical accuracy, alignment with input text). As shown in the table, the HPE significantly outperforms the GPT model, even on out-of-domain videos. This highlights that the **HF-Embodied Dataset** has indeed trained the score model to capture the nuances of video quality, particularly in terms of physical realism, which the GPT model, not trained on similar data, fails to grasp.
>
> ## Qualitative Results
>
> Furthermore, the examples in **Figure 7** illustrate the effectiveness of the HPE, showing that it produces scores that align with human preferences. We checked the response from GPT that evaluates the same videos, as shown in the following table. The scores assigned by GPT in AD and ARM scenarios are highly converse to human perceptual preference. This discrepancy underscores the importance of the **HF-Embodied Dataset**, as it provides the necessary training data for the score model to learn the correct distribution of videos that align with physical rules and those that do not. This leads to more accurate scoring that can effectively evaluate the quality of World Simulators.
>
> In summary, the **HF-Embodied Dataset** plays a vital role in training a video understanding model capable of scoring generated videos in alignment with human preferences, particularly in terms of physical accuracy. The evaluation results clearly demonstrate that the model trained on this dataset outperforms the GPT baseline, and the data itself provides a foundation for robust evaluation of World Simulators.
>
>
>
> |              |             | Video Left              |              |             | Video Right             |              |
> |--------------|-------------------------|-------------------------|-------------------------|-------------------------|-------------------------|-------------------------|
> |                  | Human   | HPE(ours)               | GPT-4o                  | Human                   | HPE(ours)               | GPT-4o                  |
> | **MC**       | EI:5 IA:5               | EI:5 IA:5               | EI:4 IA:5               | EI:5 IA:5               | EI:3 IA:5               | EI:5 IA:5               |
> | **AD**       | KE:1 SF:1 PV:2          | KE:1 SF:1 PV:1          | KE:3 SF:4 PV:4          | KE:5 SF:4 PV:5          | KE:4 SF:5 PV:5          | KE:3 SF:1 PV:3          |
> | **ARM**      | PV:5 TJ:5 EI:3 IA:1     | PV:5 TJ:5 EI:4 IA:1     | PV:4 TJ:2 EI:4 IA:3     | PV:5 TJ:4 EI:3 IA:1     | PV:5 TJ:3 EI:3 IA:1     | PV:4 TJ:1 EI:4 IA:4     |
>
>
> Thank you for your valuable time and effort in reviewing our work. We would greatly appreciate receiving your feedback on our response to facilitate further discussion. If any aspects of our explanation are unclear, please feel free to let us know. Thank you once again for your invaluable comments and consideration, which are greatly beneficial in improving our paper.

---

> ### Author Response · Authors · 2024-12-01
> **Official Comment by Authors**
>
> If our experiments and explanations as well as our **Discussion of Category** and **Justification for Dual Evaluation Framework and HF-Embodied Dataset** have alleviated your concerns. If our polishing of the paper's presentation meets your expectations, we kindly request you to consider raising your **rating**. Should you have any other doubts or dissatisfaction with the presentation of the paper, we are eager to discuss and rectify them at the earliest opportunity.

---

### Official Review · Reviewer_aqGU · 2024-11-04

**Soundness:** 3
**Presentation:** 2
**Contribution:** 3
**Rating:** 6
**Confidence:** 4

**Summary:**

The paper presents an evaluation of large scale video predictive models, also referred in the literature as world simulators. It first claims to categorize based on “degree of embodiment” or output modality, like text, images or actions. They divide this framework into a perceptual evaluation, and an embodied or manipulative one. The former uses a model trained on human collected data to score the outputs, whereas the latter, puts it in a loop with a trained video to action model and evaluates on task success.

**Strengths:**

- Its important and timely to categorize the contribution of world simulators, which makes this work useful.
- Using an HF-embodied dataset to learn a model of human preference, seems more scalable than actual human evaluation, although it might need to be updated frequently enough or kept hidden to avoid data contamination.
- In my understanding, the core contribution of this work is two fold: 1) the HF embodied dataset that tries to score the visual perceptual quality, based on human feedback. 2) Doing a closed loop evaluation by producing actions based on the predicted video.
- The idea of using world simulators as an evaluation, seems similar to line of work 1x has been doing on world modelling (https://github.com/1x-technologies/1xgpt).  It maybe useful to discuss this perspective. Just a comment, doesn't take anything away from the contribution of the work.

**Weaknesses:**

- This paper has a lot of jargon which sometimes makes it a bit hard to follow.
- The term of degree of embodiment has been used quite vaguely, which if I understand correctly is meant to be the output modality, whether its text, images or videos. Presenting it that way would be more straightforward and easy to understand.
- Embodiment has also been used very again in the Section 4.1.1 to define as an axis of evaluation for perceptual quality like obeying physical laws, which is different from the degree of embodiment term used before.

**Questions:**

- Given the hierarchical evaluation, and various axes of variation in embodiment, did the perceptual evaluation show any insights that existing benchmarks at s2 level did not? I was unable to point out a clear comparison between these and previous benchmarks, besides the HF-embodied dataset. I understand that it adds various other layers of evaluation like perspectivity and speed, but what aspects of it went unreflected previous benchmarks that this one addresses.

---

> ### Author Response · Authors · 2024-11-21
> **Response to aqGU (1/2)**
>
> We sincerely thank all reviewers for their constructive and valuable feedback on our paper.
>
> In the individual replies, we address your concerns.
>
> ### List of changes in the manuscript:
> 1. Line 81 We have fixed the ambiguity.
> 2. Line 83 We have fixed the ambiguity.
> 3. Line 93 We have fixed the ambiguity.
> 4. Line 94 We have fixed the ambiguity.
> 5. Line 135 We have fixed the ambiguity.
> 6. Line 178 We have fixed the ambiguity.
>
> ---
>
> ### Strengths
>
> **Discussion of 1x:**
>
> Thank you for your insightful suggestions. We have investigated the 1x Challenge and reflected on it from various perspectives.
>
> **1x is a fascinating work that explores the potential of autoregressive video generation models in robotics.**
>
> - **Task Setting:**
>   The 1x Challenge defines its task as follows: “Each example is a sequence of 16 first-person images from the robot at 2Hz (so 8 seconds total), and your goal is to predict the next image given the previous ones.” It focuses on the unconditioned video prediction capability of models without additional textual control. Models under this paradigm are typically fine-tuned on downstream tasks to accomplish specific robotic tasks in specific scenarios.
>
> - **Evaluation:**
>   The two evaluation approaches proposed by 1x compare the generated images with ground truth images at both the video and feature levels using traditional score-based evaluation metrics (e.g., LPIPS). Their planned action-level evaluation uses an open-loop evaluation strategy, where the predicted images are converted into actions and compared with ground truth actions through a loss function.
>
> - **Scenarios:**
>   The challenge adopts a first-person view task setting to explore these capabilities.
>
> ### In Comparison with Our Work:
>
> - **Task Setting:**
>   - Our task is to predict future videos based on instructions and real-time observations, aiming to evaluate the capability of video generation models as world simulators. This involves generating future videos that conform to physical rules and logical consistency based on given instructions.
>   - Unlike 1x, our setting explicitly incorporates text-conditioned video prediction, where the model generates videos guided by task instructions. This enables broader applications, such as action transformation to accomplish specific robotic tasks.
>
> - **Evaluation:**
>   - We aim to overcome the limitations of traditional video evaluation methods, which cannot adequately assess attributes unique to world simulators, such as physical rules and logic.
>   - Instead of score-based metrics, we employ:
>     1. **Explicit Perceptual Evaluation:** This directly evaluates attributes like physics adherence and logical consistency.
>     2. **Implicit Manipulative Evaluation:** This closed-loop evaluation maps specific attributes (e.g., trajectory quality, speed) to embodied tasks requiring those attributes. For example, robotic arm manipulation relies on high-quality trajectory generation, enabling implicit evaluation of these physical properties.
>
> - **Scenarios:**
>   - We use three distinct embodied intelligence scenarios for evaluation, encompassing both third-person and first-person view task settings.
>
> **1x is indeed an inspiring work, and while there are differences in focus between their work and ours, the discussions have provided valuable insights.**
>
> ---

---

> ### Author Response · Authors · 2024-11-21
> **Response to aqGU (2/2)**
>
> ### Weakness 1: A lot of jargon.
> **A:**  Thank you for pointing out that some parts of the paper may contain jargon, which could affect readability. We acknowledge that technical terms can make the content challenging for readers. To address this, we have revised key sections to simplify terminology and provide additional explanations for critical concepts. For example, we clarified terms such as "degree of embodiment" and contextualized them with examples to ensure accessibility. We hope these revisions improve the clarity and comprehensibility of the paper.
>
> ---
>
> ### Weakness 2: Ambiguity of "Embodiment".
> **A:**  Thank you for your valuable feedback regarding the term "degree of embodiment." We recognize the importance of precise terminology and have revised the manuscript accordingly. Specifically, we have clarified that "degree of embodiment" refers to the output modality (e.g., text, images, or videos) and have rephrased the relevant sections to present it in a more straightforward and intuitive manner. We hope this modification addresses your concern and enhances the paper's clarity.
>
> ---
>
> ### Weakness 3: Inconsistent meaning of "Embodiment".
> **A:**   Thank you for pointing out the potential confusion in our use of the term "embodiment." We have revised the manuscript to ensure consistency and clarity. Specifically, we have aligned the terminology used in Section 4.1.1 with the revised definition of "degree of embodiment" as suggested earlier. By doing so, we have eliminated any ambiguity and ensured that the term is used consistently throughout the paper. We appreciate your feedback, which has significantly helped improve the coherence of our work.
>
> ---
>
> ### Question: Detail Comparison with benchmarks at the s2 level.
> **A:**   Thank you for your thoughtful feedback. We understand your concern regarding the comparison between our evaluation approach and existing benchmarks.
>
> Past works at s2 have primarily focused on video quality and the relevance between videos and text, with evaluation axes such as background consistency, video aesthetics, and how well the video reflects the content of the textual description. These evaluations mainly address sensory coherence and visual quality. However, we introduce an evaluation framework that emphasizes attributes closely related to embodiment. For instance, in the robotic arm scenario, we assess whether the arm's interaction with the object adheres to physical constraints. Specifically, grasping rigid objects, such as wooden blocks, should not lead to deformation, while grasping flexible objects, such as sponges, should produce realistic and appropriate deformation. Similarly, in open embodied environments, we evaluate whether the speed of characters adapts to environmental changes, such as walking slower in water.
>
> Through our **Explicit Perceptual Evaluation**, we reveal key issues in current T2V models. Although these models generate visually coherent videos that perform well in previous benchmarks, they fail to demonstrate capabilities such as perceiving trajectories or understanding attributes like speed and material properties. These shortcomings, which are not reflected in existing sensory-focused benchmarks, are highlighted by our evaluation framework, which emphasizes the need for more complex embodied interaction and physical world understanding in video generation.

---

> ### Author Response · Authors · 2024-11-25
> **Official Comment by Authors**
>
> We hope this message finds you well.   We have noted the deadline for open discussion of ICLR 2025 is approaching, yet we have not yet received any feedback from you.   In light of this, we sincerely wish to know if we can receive any updated comments regarding to our submission 873, titled "WorldSimBench: Towards Video Generation Models as World Simulators".     We are very pleased to hear from you on the reviewer’s comments.

---

> ### Author Response · Authors · 2024-12-01
> **Official Comment by Authors**
>
> If our experiments and explanations as well as our discussion on the 1x and implications of our contributions, have alleviated your concerns. If our polishing of the paper's presentation meets your expectations, we kindly request you to consider raising your **rating**. Should you have any other doubts or dissatisfaction with the presentation of the paper, we are eager to discuss and rectify them at the earliest opportunity.

---

### Author Response · Authors · 2024-11-21
**Overall Response**

We sincerely thank all reviewers for their constructive and valuable feedback on our paper.

### In this post:
1. We summarize the strengths of our paper from the reviewers.
2. We summarize the changes to the updated PDF document.

In the individual replies, we address other comments.

### (1) Strengths of Our Paper

### Sound Motivation
- **aqGU**: *"It's important and timely to categorize the contribution of world simulators, which makes this work useful."*
- **kNpR**: *"This paper provides a solid overview of the world simulator."*

### Meaningful Dataset
- **aqGU**: *"Using an HF-embodied dataset to learn a model of human preference, seems more scalable than actual human evaluation."*
- **HLX1**: *"Detailed human feedback dataset (HF-Embodied Dataset) provides multi-dimensional evaluation criteria for realism and task alignment."*
- **jqzP**:
  - *"The use of human preference evaluation and fine-grained feedback allows for a more intuitive and accurate reflection of the quality and characteristics of the generated videos, including their adherence to physical rules."*
  - *"Open-Source Dataset: The HF-Embodied Dataset is open-source and can be used for various applications beyond evaluating World Simulators, such as alignment and other video generation tasks."*
- **kNpR**: *"The dataset construction involves the introduction of costly human preference annotations, which adds meaningful data to the study."*

### Solid Evaluation
- **HLX1**: *"Comprehensive dual evaluation framework assessing models on both visual quality and action-level performance."*
- **jqzP**: *"Comprehensive Evaluation: WorldSimBench provides a holistic assessment of World Simulators by considering both visual and action aspects, addressing the limitations of existing benchmarks that focus only on aesthetic quality or task completion."*
- **kNpR**: *"The experimental design is quite comprehensive, covering a wide range of aspects."*

### (2) Changes to PDF

We have proofread the paper and added extra experimental results in the revised version (**highlighted in blue**).

### Main Text
Only small fixes and wording improvements:
- **aqGU**: (Line 81) We have fixed the ambiguity.
- **aqGU**: (Line 83) We have fixed the ambiguity.
- **aqGU**: (Line 93) We have fixed the ambiguity.
- **aqGU**: (Line 94) We have fixed the ambiguity.
- **aqGU**: (Line 135) We have fixed the ambiguity.
- **aqGU**: (Line 178) We have fixed the ambiguity.

### Appendix
Additional experiments, analyses, and discussions have been incorporated in response to the reviewers' suggestions:
- **jqzP**: (Section G) Discussions about Video Generation Model and VLA.
- **HLX1, kNpR**: (Section B.1) Discussions about the future work of HF-Embodied dataset.
- **HLX1**: (Section C.2) Qualitative Results and Analyses.
- **kNpR**: (Section F.3) Evaluation with 100-trajectories and comparison with UniPi.

---

### Meta-Review · Area_Chair_iFZX · 2024-12-25

**Metareview:**

This paper introduced WorldSimBench, a new framework for evaluating world simulators. The reviewers expressed concerns about the paper's limited technical novelty, narrow scope, and aspects of its presentation. While the paper shows promise, it is not recommended for acceptance in its current form. The authors are encouraged to address the reviewers' feedback and refine the work for submission to other venues.

**Additional Comments On Reviewer Discussion:**

The paper shows great potential, but its presentation and results analysis require significant improvement.

---

### Decision · Program_Chairs · 2025-01-22

Reject